# Thermo-Logic Attention Dynamics (TLAD): Differentiable Hard Logic via Disentangled Lagrangian Mechanics

## Abstract

Neuro-symbolic (NeSy) AI aims to integrate the perceptual capabilities of neural networks with symbolic reasoning to solve complex tasks requiring interpretable and structured inference. However, existing approaches face a persistent trilemma: they struggle to simultaneously achieve end-to-end differentiability, strict hard-constraint satisfaction, and high data efficiency. Inspired by thermodynamic systems that spontaneously evolve toward stable states through energy minimization, we propose Thermo-Logic Attention Dynamics (TLAD). At its core, TLAD reformulates discrete symbolic constraints as differentiable energy potentials defined over a continuous attention state space. A topologically disentangled attention mechanism guides the system's evolution along this energy landscape, driving it to relax into a stable, constraint-satisfying solution. Empirically, TLAD exhibits top-tier performance on complex constrained tasks such as Sudoku (combinatorial logic) and maintains high accuracy under extremely low-data regimes, where existing baselines catastrophically fail. Furthermore, we investigate the boundary of neuro-symbolic models on sparse-constraint, high-freedom problems like maze navigation. These results demonstrate that thermodynamically grounded inductive biases provide a principled resolution to the fundamental trilemma of NeSy integration.

## 1. Introduction

Neuro-Symbolic (NeSy) AI aims to merge neural perception with symbolic reasoning (d'Avila Garcez & Lamb, 2020; Gibaut et al., 2023; Feldstein et al., 2024; Colelough & Regli, 2025), but is hindered by a "Correct-

ness–Efficiency–Differentiability" trilemma. The core issue: logical constraints are treated as extrinsic supervisors, rather than as intrinsic properties of the model's representational dynamics.

This manifests in three dominant paradigms, each compromising one vertex of the trilemma.

Regularization-based methods retain efficiency and differentiability by encoding logical knowledge as continuous surrogates but sacrifice correctness. Semantic Loss (Xu et al., 2018) maps propositional constraints to a probabilistic penalty for neural outputs. These relaxations inherently lead to residual violations, where even converged predictions may breach hard domain rules. MultiplexNet(Hoernle et al., 2021) hard-wires disjunctive normal form constraints into the output layer for exact satisfaction, but this guarantee fails when expert rules are incomplete or brittle, such as in occluded real-world perceptual tasks.

Solver-integrated architectures enforce hard constraints by embedding optimization engines while sacrificing efficiency for correctness. OptNet(Amos & Kolter, 2021) and CVXPY Layers(Agrawal et al., 2019) embed convex solvers into networks and compute gradients via implicit KKT differentiation. They are rigorous but scale poorly and struggle with the discrete, non-convex nature of combinatorial logic. SATNet(Wang et al., 2019) approximates Boolean satisfiability through semidefinite relaxation, yet exact solvers still impose prohibitive latency for real-time use.

Iterative NeSy frameworks alternate between neural perception and symbolic reasoning to refine predictions while sacrificing end-to-end differentiability for improved correctness. Abductive Learning (ABL)(Zhou & Huang, 2021) treats symbolic rules as consistency supervisors in an Expectation-Maximization loop, using abduced pseudo-labels to retrain the perception module. Abductive Reflection (Abl-RefL)(Hu et al., 2025) enhances this with multi-round self-correction, while NSCL(Mao et al., 2019) decouples reasoning via executable programs and biased gradient estimators like Gumbel-Softmax to bridge non-differentiable gaps. These systems incur heavy computational costs and prevent neural models from internalizing reasoning as an intrinsic property.

[1] Anonymous Institution, Anonymous City, Anonymous Region, Anonymous Country. Correspondence to: Anonymous Author <anon.email@domain.com>.

Preliminary work. Under review by the International Conference on Machine Learning (ICML). Do not distribute.

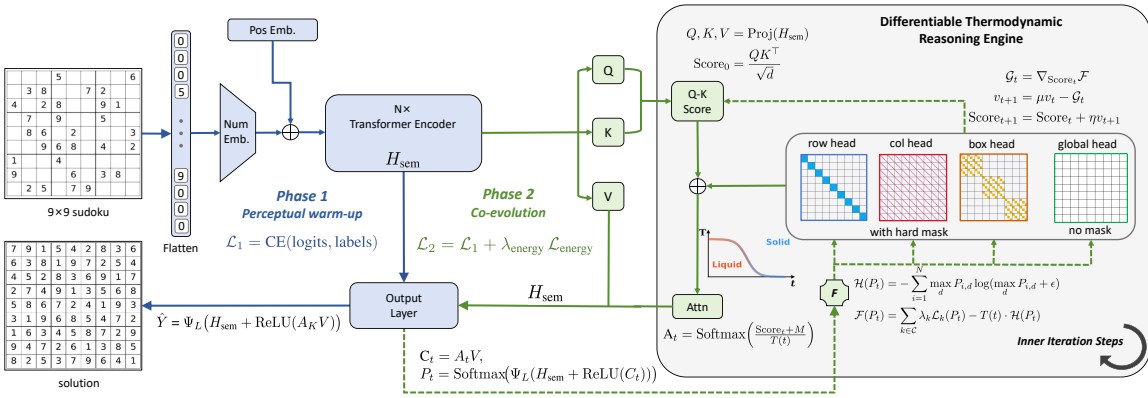

*Figure 1.* **TLAD Framework Overview (Dual-System Paradigm). Phase 1 (Left):** Transformer Encoder generates initial semantic embeddings (System 1, perceptual warm-up). **Phase 2 (Right):** Differentiable Thermodynamic Reasoning Engine (System 2, physical attention evolution). Key Innovations: (1) Topologically Disentangled Heads (Row/Col/Box) enforce hard constraints via masks; (2) Hamiltonian Dynamics (top equations) add momentum for energy landscape traversal; (3) Thermodynamic Annealing ($T$) modulates entropy, driving a "Liquid (exploration) $\rightarrow$ Solid (crystallized)" phase transition.

To resolve this, we propose **Thermo-Logic Attention Dynamics (TLAD)**, a framework that mitigates the geometric-topological mismatch by reformulating reasoning as a *differentiable energy minimization process*. Drawing inspiration from thermodynamic models (Chang, 2026; Shiratori et al., 2025; Nateghi et al., 2025) and Lagrangian networks (Lutter et al., 2019; Schulze et al., 2025; Diaz et al., 2024), TLAD internalizes logical rules as *intrinsic energy potentials*. By employing a disentangled attention topology driven by Lagrangian forces, our method constructs a smooth path from the continuous neural space to the discrete logical manifold. This design enables strict enforcement of hard constraints while maintaining a rich, informative gradient flow, offering a rigorous and efficient pathway toward reliable NeSy integration.

Contributions. Our work makes three foundational contributions to neuro-symbolic AI: 1) Methodological Innovation: We propose TLAD, a thermodynamically grounded framework that resolves the NeSy trilemma by formalizing reasoning as a controlled phase transition, with Topologically Disentangled Attention as its backbone to eliminate gradient conflict and enforce hard constraints, 2) Empirical Validation: TLAD achieves state-of-the-art performance across data scales and task types, demonstrating superior data efficiency, rapid error correction, and robust physics-inspired inductive bias, 3) Theoretical Insight: We clarify the operational boundary of NeSy reasoning. Its utility is task-dependent, excelling in dense-constraint or sparse-data/high-complexity regimes while offering no advantage in simple sparse-constraint tasks with abundant data.

## 2. Related Work

NeSy AI aims to bridge continuous learning and discrete reasoning. Existing NeSy reasoning methods mainly fall into three categories: soft regularization methods (Xu et al., 2018; Marc Fischer, 2019; Hoernle et al., 2021), which struggle with residual constraint violations; solver-integrated architectures (Amos & Kolter, 2021; Agrawal et al., 2019; Wang et al., 2019; Dragone et al., 2021), which suffer from poor scalability or high latency; and iterative NeSy frameworks (Zhou & Huang, 2021; Hu et al., 2025; Mao et al., 2019), which incur heavy computational costs and fail to internalize reasoning. Meanwhile, thermodynamic and Lagrangian models have attracted growing attention in machine learning for their advantages in interpretability, stability and constrained optimization, yet their integration for discrete symbolic reasoning remains underexplored

In recent years, thermodynamic models have attracted extensive attention in machine learning, as they can endow data-driven models with physical interpretability and stability by embedding thermodynamic principles. Relevant studies have applied thermodynamics to model reasoning analysis, generalization improvement, and performance evaluation (Chang, 2026; Shiratori et al., 2025; Kodama & Hinczewski, 2025), but few focus on discrete symbolic reasoning tasks. Meanwhile, Lagrangian models have become a research hotspot due to their advantage in handling constrained optimization problems by encoding constraint information into neural networks. Existing works have combined Lagrangian methods with deep learning for constraint adaptation and system stability improvement (Zheng & Fu, 2025; Diaz et al., 2024; Wang et al., 2024), yet their integration with thermodynamics for discrete reasoning is rarely explored.

## 3. Thermo-Logic Attention Dynamics

We frame NeSy reasoning as a dynamic phase transition (not static inference). Unlike standard Transformers, **TLAD** models attention as a physical system evolving to minimize thermodynamic free energy—enforcing hard logical constraints via Hamiltonian-inspired dynamics while maintaining perceptual grounding through an **Implicit Semantic Anchor**. See Figure 1 for a detailed architectural overview.

### 3.1. Thermodynamic State Space and Relaxation

Given perceptual input $X$ (e.g., visual Sudoku), we infer a symbolic state $Y$ satisfying hard constraints $\mathcal{C}$. For end-to-end differentiability, we relax discrete logic into a continuous probability manifold.

Let $H_{\text{sem}} \in \mathbb{R}^{N \times d}$ be the perceptual encoder's semantic embedding (System 1). The reasoning state is a dynamic **Attention Score Tensor** $S_t \in \mathbb{R}^{H \times N \times N}$ (unbounded logit space). The observable distribution $P_t$ is derived via:

$$
\begin{aligned}
A_t &= \text{Softmax}\left(\frac{S_t + M}{T_t}\right), \quad C_t = A_t V, \\
P_t &= \text{Softmax}\left(\Psi_L\left(H_{\text{sem}} + \text{ReLU}(C_t)\right)\right) \in \Delta^{N \times D},
\end{aligned} \tag{1}
$$

where $M$ (topological mask, Section 3.3), $T_t$ (time-dependent temperature), and $\Psi_L$ (learnable logit head). The residual form $H_{\text{sem}} + \text{ReLU}(C_t)$ enforces the **Implicit Semantic Anchor**: weak/unstable $C_t$ defaults to $H_{\text{sem}}$ for robustness (no explicit regularization). $S_t$ is the dynamic logit-level tensor, $P_t$ the symbolic probability distribution for decisions.

### 3.2. The Thermo-Logic Energy Landscape

System evolution is governed by time-dependent free energy $\mathcal{F}$, with two competing potentials (logical order + phase transition regulation). We define them separately then combine:

$$
\mathcal{F}_{\text{lag}}(P_t, \boldsymbol{\lambda}) = \underbrace{\sum_{k \in \mathcal{C}} \lambda_k \mathcal{L}_k(P_t)}_{\text{Lagrangian Potential (Logic)}} \tag{2}
$$

$$
\mathcal{F}_{\text{ent}}(P_t, T_t) = \underbrace{T_t \cdot \mathcal{H}(P_t)}_{\text{Entropic Potential (Exploration)}} \tag{3}
$$

$$
\mathcal{F}(P_t, \boldsymbol{\lambda}, T_t) = \mathcal{F}_{\text{lag}} - \mathcal{F}_{\text{ent}} \tag{4}
$$

where: - $\mathcal{F}_{\text{lag}}$ (Eq. 2): Enforces hard logical constraints - $\mathcal{F}_{\text{ent}}$ (Eq. 3): Regulates exploration-exploitation tradeoff

**Lagrangian Potential (Hard Logic).** Each $k \in \mathcal{C}$ maps to a differentiable penalty $\mathcal{L}_k \geq 0$. Lagrange multipliers $\lambda_k$ (dynamic spring constants) stiffen via dual ascent. For Sudoku:

- **Row/Col/Box Uniqueness:**

$$
\mathcal{L}_{\text{row},r} = \sum_{d=1}^{9} \left(\sum_{c=1}^{9} P_{(r,c),d} - 1\right)^2 \tag{5}
$$

- **Cell Validity:**

$$
\mathcal{L}_{\text{cell},i} = 1 - \sum_{d=1}^{9} P_{i,d}^2 \tag{6}
$$

(encourages one-hot decisions for consistency)

As $\lambda_k \to \infty$, $\mathcal{F}$'s global minimum aligns with the feasible logical manifold.

**Entropic Potential.** We use **modal entropy**:

$$
\mathcal{H}(P_t) = -\sum_{i=1}^{N} \max_d P_{i,d} \log\left(\max_d P_{i,d} + \epsilon\right), \tag{7}
$$

where $\epsilon > 0$ avoids log-singularities. Measuring dominant prediction uncertainty, $T_t$ drives a "liquid (exploration) → solid (crystallization)" phase transition, solving the combinatorial problem via physical relaxation.

### 3.3. Topologically Disentangled Attention

Standard dense attention has gradient conflicts for orthogonal constraints. **Topologically Disentangled Attention** factorizes Hamiltonian forces into independent subspaces.

We partition $H$ attention heads into groups, each with a hard structural mask $M^{(h)} \in \{0, -\infty\}$ (constraint topology). For Sudoku, four "inference lanes":

- **Row Heads:** $M_{ij}^{(h)} = 0$ iff $i, j$ share a row.

- **Column Heads:** $M_{ij}^{(h)} = 0$ iff $i, j$ share a column.

- **Box Heads:** $M_{ij}^{(h)} = 0$ iff $i, j$ lie in the same $3 \times 3$ box.

- **Global Heads:** Unmasked (captures cross-topology dependencies).

This isolates constraint propagation, avoiding interference between orthogonal rules.

### 3.4. Hamiltonian-Inspired Inference Dynamics

Inference uses momentum-augmented updates on $S_t$, approximating discretized Hamiltonian flow:

$$
\begin{aligned}
\mathcal{G}_t &= \text{SurgicalRoute}\left(\nabla_{S_t} \mathcal{F}, \boldsymbol{\lambda}\right) \\
v_{t+1} &= \mu v_t - \mathcal{G}_t \\
S_{t+1} &= S_t + \eta v_{t+1}
\end{aligned} \tag{8}
$$

**Algorithm 1** TLAD Forward Inference Dynamics

**Require:** Input $X$, Perception $\Phi$, LogitHead $\Psi_L$, Steps $K$, Masks $M$
**Ensure:** Final logits $\hat{Y}$
1: // **Phase 1: Perceptual Initialization**
2: $H_{\text{sem}} \leftarrow \Phi(X)$
3: $Q, K, V \leftarrow \text{Proj}(H_{\text{sem}}); \quad S_0 \leftarrow QK^\top/\sqrt{d}$
4: // **Phase 2: Thermodynamic Evolution**
5: **for** $t = 0 \dots K-1$ **do**
6: $\quad T_t \leftarrow \text{Anneal}(t)$ {Cooling schedule}
7: $\quad A_t \leftarrow \text{Softmax}\big((S_t + M)/T_t\big)$
8: $\quad P_t \leftarrow \text{Softmax}\big(\Psi_L\big(H_{\text{sem}} + \text{ReLU}(A_t V)\big)\big)$
9: $\quad \mathcal{F}_{\text{lag},t} \leftarrow \sum \lambda_k \mathcal{L}_k(P_t)$ {Lagrangian Potential}
10: $\quad \mathcal{F}_{\text{ent},t} \leftarrow T_t \mathcal{H}(P_t)$ {Entropic Potential}
11: $\quad \mathcal{F}_t \leftarrow \mathcal{F}_{\text{lag},t} - \mathcal{F}_{\text{ent},t}$ {Total Free Energy}
12: $\quad \mathcal{G} \leftarrow \nabla_{S_t} \mathcal{F}_t$ {Backprop through readout}
13: $\quad \mathcal{G} \leftarrow \text{SurgicalRoute}(\mathcal{G}, \boldsymbol{\lambda})$
14: $\quad V_{t+1} \leftarrow \mu V_t - \mathcal{G}$ {Hamiltonian update}
15: $\quad S_{t+1} \leftarrow S_t + \eta V_{t+1}$
16: **end for**
17: // **Output final logits**
18: $A_K \leftarrow \text{Softmax}\big((S_K + M)/T_K\big)$
19: $\hat{Y} \leftarrow \Psi_L\big(H_{\text{sem}} + \text{ReLU}(A_K V)\big) \hat{Y}$

---

Momentum $v_t$ stabilizes trajectories, preventing oscillation during phase transition.

**Surgical Gradient Routing.** Backpropagate $\nabla_{P_t} \mathcal{F}$ through Eq. 1 to get $\nabla_{S_t} \mathcal{F}$. Each head $h$'s gradient is scaled by its constraint type's $\lambda$:

$$\mathcal{G}^{(h)} \leftarrow \lambda_{\text{type}(h)} \cdot \mathcal{G}^{(h)}, \tag{9}$$

ensuring force-inference lane alignment.

### 3.5. Dual-Timescale Optimization

We employ a bilevel optimization strategy to reconcile statistical learning with logical rigidity:

- **Slow Timescale ($\theta$):** Neural parameters (encoder, $\Psi_L$) are updated slowly ($\eta_{\text{slow}}$) to shape a robust energy landscape.

- **Fast Timescale ($\boldsymbol{\lambda}$):** Lagrange multipliers are updated rapidly ($\eta_{\text{fast}} \gg \eta_{\text{slow}}$).

This separation ensures logical constraints are "hardened" into infinite potential barriers faster than perception adapts, forcing the network to internalize valid reasoning patterns rather than memorizing data shortcuts.

## 4. Sudoku Experiments

In this section, we evaluate our framework on the Sudoku reasoning task. We aim to answer four refined research questions:

- **Q1 (Performance in Data-Rich Regimes):** In data-sufficient settings (80k training examples), can TLAD achieve accuracy comparable to specialized solvers such as RRN while using only a continuous thermodynamic reasoning mechanism?

- **Q2 (Data Efficiency):** Does TLAD exhibit significantly higher sample efficiency than neural or NeSy baselines, maintaining strong performance even with extremely limited training data (1k–2k examples)?

- **Q3 (Robustness under Dense Constraints):** In highly constrained, low-degeneracy tasks like Sudoku, does TLAD demonstrate robust generalization in small-data regimes?

- **Q4 (Training Efficiency):** Does TLAD converge faster or require fewer training epochs than competing models?

All experiments were conducted on a high-performance workstation equipped with a single NVIDIA RTX 4090 GPU (24GB) and an Intel Xeon Gold 6430 CPU (16 vCPUs). We report the peak test accuracy during training, along with the time (minutes) and epoch at which this accuracy was first attained. To account for stochastic variability in the 1k sample regime, we evaluate each model under five independent runs with different random seeds.

### 4.1. Dataset

We evaluate on the *Big Kaggle Sudoku* dataset, a subset of a bigger dataset containing 1 million boards hosted on Kaggle, which is also used in DDReasoner(Zhang et al., 2025). To benchmark data efficiency, we curated five subsets with total sizes of 100k, 25k, 6.25k, 2.5k, and 1.25k. Adopting an 8:1:1 split (80% training, 10% validation, 10% testing), these yield effective training set sizes of 80k, 20k, 5k, 2k, and 1k.

### 4.2. Comparison Methods

We select three representative baselines to isolate the thermodynamic reasoning engine's contribution: 1) **Vanilla Transformer**: A standard Transformer without specialized reasoning modules, sharing TLAD's System 1 structure and parameters. 2) **SATNet**(Wang et al., 2019): A NeSy framework integrating a differentiable MaxSAT solver for reasoning. 3) **RRN**(Palm et al., 2018): A cutting-edge all-neural approach optimized for Sudoku. Detailed hyperparameters are provided in AppendixB.

*Table 1.* **Main Results on Sudoku Benchmark.** Comparison across data regimes. TLAD achieves superior data efficiency: with only **1k training samples**, it attains **77.3%** accuracy (average over five runs), while all baselines collapse to 0%. It also converges faster on larger datasets.

| MODEL | TEST ACCURACY (%) | | | | | TRAINING TIME (MIN/EPOCH) | | | |
|---|---|---|---|---|---|---|---|---|---|
| | **80K** | **20K** | **5K** | **2K** | **1K**[†] | **80K** | **20K** | **5K** | **2K** |
| VANILLA TRANS. | 87.8 | 77.6 | 28.3 | 2.4 | 0.0 | 24.0 (93) | **6.5 (100)** | 1.4 (83) | **0.7 (66)** |
| SATNET | 96.4 | 95.5 | 85.4 | 14.8 | 0.0 | 385.0 (61) | 130.4 (87) | 28.4 (82) | 14.8 (97) |
| RRN | **99.8** | **99.8** | 93.4 | 35.8 | 0.0 | 11.8 (24) | 8.1 (76) | 3.0 (100) | 1.2 (100) |
| **TLAD (OURS)** | **99.8** | 98.9 | **94.0** | **84.1** | **77.3** | **10.0 (13)** | 7.1 (21) | **1.1 (15)** | 3.3 (59) |

[†]TLAD results on 1k samples are averaged over five independent runs (77.3±5.9%). All baselines failed to converge (0% accuracy) on this extremely low-data regime. Training time for 1k is omitted as baselines did not reach meaningful accuracy.

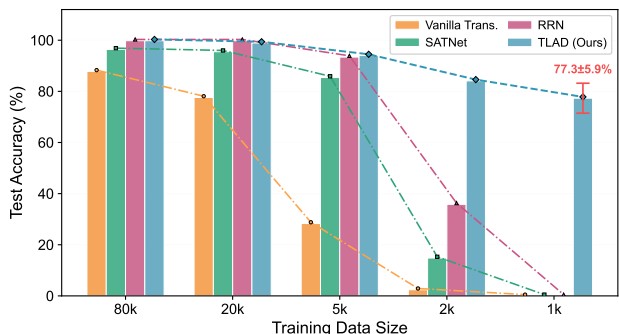

*Figure 2.* **Data Efficiency on Sudoku.** Test accuracy across training data sizes. TLAD maintains 77.3% accuracy (five runs, ±5.9% std) at 1k samples while all baselines reach 0%. At 2k samples, TLAD reaches 84.1% accuracy, far exceeding RRN (35.8%), SAT-Net (14.8%), and Vanilla Transformer (2.4%). Shaded region indicates standard deviation, demonstrating TLAD's robustness in extreme data scarcity.

### 4.3. Main Results and Analysis

As shown in Table 1 and Figure 2, TLAD demonstrates compelling advantages in data efficiency and training speed across all regimes.

**Q1: Performance in Data-Rich Regimes.** In data-rich settings (80k samples), TLAD achieves 99.8% test accuracy, matching RRN (99.8%) and outperforming SATNet (96.4%) and Vanilla Transformer (87.8%). Unlike RRN with explicit graph message passing and SATNet with hard SDP solvers, TLAD adopts only continuous thermodynamic reasoning yet enforces global Sudoku constraints with near-perfect precision. Q1 is resolved: sufficient data enables TLAD to rival specialized discrete solvers via energy-based symbolic grounding.

**Q2: Data Efficiency.** Supported by Table 1 and visualized in Figure 2, TLAD gains decisive advantages under data scarcity. It hits 84.1% accuracy at 2k samples, far exceeding RRN (35.8%), SATNet (14.8%) and Vanilla Transformer (2.4%). At 1k samples, all baselines reach 0% while TLAD maintains 77.3% (±5.9% std, 5 runs). This verifies TLAD leverages structural priors over statistical memorization. Q2

is supported: TLAD achieves superior sample efficiency with strong performance on extremely limited data.

**Q3: Robustness under Dense Constraints.** TLAD's low-data robustness originates from aligned energy landscape and Sudoku's logical structure. In this dense-constraint, low-freedom task, energy minimization reduces constraint violations directly. Each Hamiltonian descent step cuts row, column and box conflicts stably, providing clear optimization signals for scarce-data learning. Q3 is confirmed: TLAD realizes robust small-sample generalization in dense-constraint tasks via well-aligned inductive bias of thermodynamic dynamics.

**Q4: Training Efficiency.** Table 1 demonstrates TLAD's superior training efficiency. It converges in 10.0 min (13 epochs) on 80k samples, faster than SATNet (385.0 min, 61 epochs) and RRN (11.8 min, 24 epochs). At 5k samples, TLAD converges in 1.1 min (15 epochs), outperforming all baselines except Vanilla Transformer which trades accuracy for speed. Fast convergence stems from embedded Sudoku rules in energy functions, enabling rule-guided optimization. Q4 is validated: TLAD achieves higher accuracy with less training time via thermodynamic formulation encoding domain constraints inherently.

## 5. Mechanistic Analysis

In this section, we employ the same Sudoku benchmark introduced in Section 4 and investigate the inner workings of TLAD through three complementary lenses: the functional nature of its reasoning process, the robustness of hyperparameters across data regimes, and the necessity of its core architectural components.

### 5.1. Reasoning Nature via Step-wise Analysis

We first analyze performance variation with reasoning depth $K$ to clarify TLAD's inference mechanism and depth efficiency trade-off via two key questions.

- **Q1 (Mechanism of Integration):** Does the thermody-

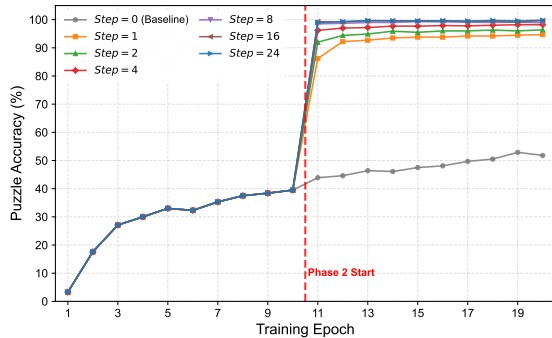

*Figure 3.* **Thermodynamic Phase Transition at Phase 2.** Phase 1 (1–10): Baseline ($K = 0$) stagnates at $\approx 39.5\%$, limited to statistical pattern matching. Phase 2 (Epoch 11): Thermodynamic engine activation enables $K = 1$ to jump accuracy to 94.7%.

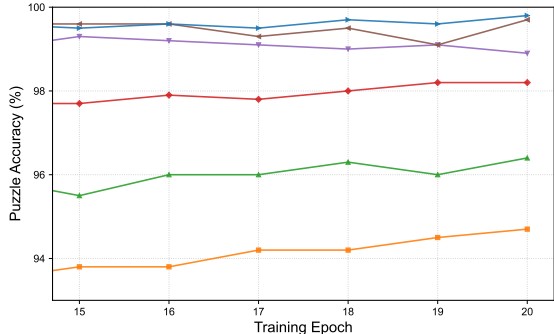

*Figure 4.* **Accuracy Convergence with Reasoning Depth.** Accuracy evolution for Epochs 15–20 (Phase 2). Increasing reasoning depth from $K = 1$ to $K = 24$ refines accuracy, reaching a peak of 99.8% with predictable computational scaling.

*Table 2.* **Efficiency–Accuracy Trade-off.** Training/inference cost vs. performance. $K = 1$ offers optimal efficiency, $K = 24$ maximizes rigor for hard cases.

| INNER STEPS ($K$) | TRAINING TIME (MIN/20 EPOCHS) | INFERENCE TIME (S/10K SAMPLES) | BEST TEST ACCURACY |
|---|---|---|---|
| 0 (BASE) | **5.8** | **1.1** | 52.9% |
| 1 | 7.1 | 1.2 | 94.7% |
| 2 | 8.0 | 1.3 | 96.4% |
| 4 | 9.8 | 2.2 | 98.2% |
| 8 | 13.4 | 3.2 | 99.3% |
| 16 | 20.5 | 6.7 | 99.7% |
| 24 (FULL) | 27.7 | 9.5 | **99.8%** |

namic engine show rapid initial convergence, indicating an energy landscape guiding solutions toward valid manifold in few updates?

- **Q2 (Depth–Efficiency Trade-off):** What is the minimal reasoning depth for near-optimal performance, and how does increasing inference steps trade computational cost for robustness on hard combinatorial instances?

**Experimental Setup** We use 80k Sudoku and train models with reasoning depth $K \in \{0, 1, 2, 4, 8, 16, 24\}$. Each configuration trains 10 epochs per phase. Detailed hyperparameters are in Appendix B.

**Q1 (Mechanism of Integration).** Supported by Table 2, Figure 3 and Figure 4, TLAD achieves 52.9% accuracy at $K = 0$ (pure perceptual Transformer), unable to enforce global Sudoku constraints despite local pattern capture. Enabling one thermodynamic step at $K = 1$ jumps accuracy to 94.7% (41.8% gain) with 0.1 s higher inference latency. This stems from the learned Hamiltonian energy landscape, whose gradient provides effective descent direction to reduce constraint violations and steer outputs toward valid configurations. Q1 is resolved: the thermodynamic engine enables rapid initial convergence via energy function align-

ment with logical constraints, allowing few updates for most correction benefits.

**Q2 (Depth–Efficiency Trade-off).** Table 2 and Figure 4 show accuracy improves with $K$: 98.2% at $K = 4$, 99.7% at $K = 16$, and 99.8% at $K = 24$. Computational cost grows near-linearly: training time 7.1–27.7 minutes, inference time 1.2–9.5 s. For $K \geq 2$, each additional step adds 0.4 s inference latency (per 10k samples) and 0.9 minutes training time (per 20 epochs) with fixed cost per iteration. Regular scaling and diminishing accuracy gains let practitioners trade compute for correctness. Q2 is resolved: TLAD's reasoning depth implements a controllable efficiency-rigor trade-off, enabling principled $K$ selection via predictable performance-compute scaling.

**5.2. Parameter Sensitivity Analysis**

We evaluate TLAD's robustness by examining four key hyperparameters model capacity, thermodynamic schedule, annealing rate, and constraint strength, and their performance impact across data regimes. This addresses two critical deployment questions:

- **Q3 (Tuning Robustness):** Do TLAD's hyperparameters exhibit wide tolerance ranges, where suboptimal choices cause only gradual degradation rather than catastrophic failure?

- **Q4 (Scale Invariance):** Are sensitivity patterns, including unimodal distributions and peak locations, consistent across data regimes, enabling parameter transfer without per-scale retuning?

**Experimental Setup.** We conduct dual regime sensitivity analysis on 80k training set for performance ceilings and 5k set for data scarcity. For each parameter, we test values spanning an order of magnitude around the base configuration while fixing others. All experiments follow the same 10

*Table 3.* Parameter Sensitivity Across Data Regimes: 80k vs 5k Training Samples

| Parameter | Value | Test Accuracy | |
| --- | --- | --- | --- |
| | | 80k Data | 5k Data |
| *I. Model Capacity (Hidden Dimension $d$)* | | | |
| Dimension $d$ | 32 | 66.1% | 0.0% |
| | 64 | 99.6% | 44.4% |
| | 128 | 99.6% | 92.6% |
| | **256 (BASE)** | **99.8%** | 94.0% |
| | 512 | **99.8%** | **96.6%** |
| *II. Thermodynamic Schedule (Transition Center $\tau$)* | | | |
| Center $\tau$ | 0.2 | **99.8%** | 90.4% |
| | 0.4 | **99.8%** | 93.7% |
| | **0.8 (BASE)** | **99.8%** | **94.0%** |
| | 1.0 | **99.8%** | 88.4% |
| *III. Annealing Slope ($\gamma$)* | | | |
| Slope $\gamma$ | 1.0 | 99.3% | 93.0% |
| | **2.0 (BASE)** | **99.8%** | **94.0%** |
| | 5.0 | 99.6% | 83.1% |
| *IV. Initial Constraint Stiffness ($\lambda_{start}$)* | | | |
| Multiplier $\lambda_{start}$ | 5.0 | 97.7% | 79.9% |
| | **10.0 (BASE)** | **99.8%** | **94.0%** |
| | 20.0 | 98.6% | 88.3% |

epochs per phase training protocol as Section 5.1. Detailed hyperparameter grids are in Appendix B.

**Q3 (Tuning Robustness).** Supported by Table 3, TLAD has forgiving hyperparameter tuning across all four parameters, with suboptimal choices causing gradual rather than catastrophic degradation. Model capacity $d$ shows threshold behavior: $d \geq 64$ achieves 99.6% on 80k data, $d \geq 128$ achieves >92% on 5k data without collapse. Thermodynamic schedule $\tau$ is highly robust, maintaining 99.8% on 80k data and a broad optimum on 5k data. Annealing rate $\gamma$ and constraint strength $\lambda_{start}$ have moderate sensitivity but retain >83% accuracy at worst tested values under scarcity. Q3 is answered: TLAD's hyperparameters have wide tolerance ranges, enabling deployment without exhaustive tuning.

**Q4 (Scale Invariance).** Table 3 shows optimal hyperparameters are stable across regimes: $\tau = 0.8$, $\gamma = 2.0$, $\lambda_{start} = 10.0$ deliver peak performance on both 80k and 5k data. Sensitivity patterns remain unimodal with coincident peaks, no optimal shift or modality change. Data scarcity only amplifies performance drops for suboptimal choices such as $\gamma = 5.0$, while optimal points stay fixed. Model capacity scales predictably under scarcity, enabling parameter transfer without per-scale retuning. Q4 is resolved: TLAD has strong cross scale consistency, with optimal parameters and unimodal patterns preserved across regimes for robust deployment.

**Mechanistic Insight.** TLAD's unimodal sensitivity stems from its free energy formulation $F = E - TS$, where each hyperparameter modulates the trade-off between constraint energy $E$ and entropy $S$ at temperature $T$. Model capacity balances representation completeness and over parameterization noise; $\tau$ aligns exploration with constraint network relaxation; $\gamma$ tunes cooling rate; $\lambda$ negotiates rule enforcement and adaptation. Each peak corresponds to minimal free energy equilibrium, reflecting physical trade-offs in the energy landscape, not arbitrary tuning.

### 5.3. Ablation Study

Having established TLAD's convergence dynamics and parameter robustness, we now dissect its architectural necessity via a systematic component ablation analysis. Key mechanistic questions regarding component criticality and phase synergy are addressed, and full details (including experimental setup, variant configurations, and quantitative results) are provided in **Appendix C**.

Briefly, ablation results confirm two core insights: (1) adaptive Lagrangian constraints are indispensable for performance, with static constraints causing catastrophic accuracy degradation; (2) the two-phase (perceptual priming + thermodynamic reasoning) design relies on synergistic coordination, with neither phase achieving meaningful performance in isolation.

## 6. Boundary Case: Sparse Constraints

In this section, we investigate the applicability boundary of NeSy models under sparse logical constraints and high solution freedom. NeSy systems including TLAD fundamentally rely on explicit logical constraints to guide learning and correct neural predictions. However, in tasks with minimal constraints and highly degenerate solution space, it remains unclear whether NeSy integration still offers meaningful advantage over pure statistical fitting.

To probe this boundary, we study maze navigation: a path planning task with only local hard constraints and exponentially large valid solution sets. Due to sparse constraint structure and high combinatorial freedom, maze navigation represents a challenging regime for constraint driven reasoning.

Through these experiments, we aim to answer three refined research questions:

- **Q1 (Performance in Simple Regimes):** In data-rich or structurally simple sparse constraint tasks (e.g., 5×5 mazes with 5k samples), can NeSy models match or exceed purely neural baselines performance?
- **Q2 (Advantage in Complex/Scarce Regimes):** In complex or data scarce settings (e.g., 10×10 mazes

with 2k samples), does NeSy inductive bias provide unique edge?

- **Q3 (Applicability Boundary):** What defines the operational boundary of NeSy reasoning?

Existing NeSy maze-solving approaches rely on architectures with strong structural inductive biases, following three dominant paradigms: reinforcement learning with policy rollouts (global reward-guided iterative trajectory refinement (Chang, 2026)), autoregressive multi-step prediction (temporally structured local move sequence modeling (Nolte et al., 2024)), and constraint-guided diffusion (logical consistency via iterative denoising (van Krieken et al., 2025)).

All decompose reasoning temporally, iteratively or via trial-and-error, transforming global sparse constraint satisfaction into local tractable decisions. Yet they introduce auxiliary inductive biases orthogonal to our core question: can NeSy systems facilitate reasoning in sparse constraint environments? We therefore use a single-pass output head for end-to-end evaluation free of task decomposition or external scaffolding.

To ensure statistical rigor, test accuracy is reported as mean ± standard deviation over independent trials with different random seeds: 5 runs for 1k and 2k samples (high variance) and 3 runs for 5k samples.

### 6.1. Dataset

We evaluate TLAD on the *Maze* dataset(Ivanitskiy et al., 2023), focusing on $5\times5$ and $10\times10$ grid sizes. For each size, we train on 5k, 2k and 1k examples: settings where baselines struggle to recover valid paths.

### 6.2. Methodology and Comparison Methods

**TLAD for maze.** For maze navigation, the structural prior is unique connected path, encoded via four geometric and topological energy constraints: wall impenetrability, endpoint inclusion, reachability restricted to input derived valid path region, degree consistency with degree one at endpoints and degree two at internal nodes. Attention heads are split into local heads (masked to 4 neighbor spatial adjacency for path continuity) and global heads (fully connected for start-end awareness). Detailed hyperparameters are in **AppendixB**.

**Comparison Methods.** We compare TLAD with Vanilla Transformer to evaluate TLAD mechanism effectiveness. The baseline is trained for 100 epochs to ensure complete convergence, providing stable performance reference.

### 6.3. Main Results and Analysis

**Q1 (Performance in Simple Regimes).** Supported by Table4, purely neural models perform strongly on $5\times5$ mazes

*Table 4.* **Maze Generalization Performance.** Test accuracy (%) under low-data regimes. Best results per setting are bolded.

| GRID SIZE | TRAIN SIZE | TEST ACCURACY (%) | |
| --- | --- | --- | --- |
| | | VANILLA TRANS. | TLAD (OURS) |
| $5 \times 5$ | 5K | **97.76**± 1.31 | 95.31± 1.85 |
| | 2K | **78.42**± 7.32 | 65.36± 7.95 |
| | 1K | 45.60± 5.25 | **63.84**± 7.44 |
| $10 \times 10$ | 5K | 23.89± 8.94 | **57.32**± 12.97 |
| | 2K | 4.16± 1.08 | **56.56**± 17.50 |
| | 1K | 0.80± 0.00 | **25.28**± 36.73 |

with 5k or 2k samples via path pattern memorization. TLAD underperforms due to insufficient discriminative symbolic signals from sparse constraints and high solution degeneracy. Q1 is answered: NeSy integration offers no benefit for simple sparse constraint tasks with sufficient data and may degrade performance via inductive overhead.

**Q2 (Advantage in Complex/Scarce Regimes).** Table4 shows purely neural models fail catastrophically on $10\times10$ mazes or 1k sample $5\times5$ mazes, lacking extrapolation capability. TLAD uses geometric and topological priors to sustain reasoning when statistical signals vanish. Q2 is answered: NeSy models provide a critical performance lower bound in sparse data, high complexity regimes despite weak constraints.

**Q3 (Applicability Boundary).** TLAD has higher variance in challenging regimes (e.g., 25.28±36.73% for $10\times10$ mazes with 1k samples, Table4), evidence of active reasoning not instability. Unlike baselines (near zero accuracy, low variance, generalization collapse), TLAD attempts global constraint satisfaction. Small variations affect convergence in fragile valid solution spaces, with high variance reflecting reasoning potential under uncertainty. Q3 is answered: NeSy utility is task dependent—boosting accuracy in strongly constrained domains like Sudoku, while its maze advantage lies in sample efficiency and reasoning feasibility under complexity or data scarcity, not peak performance.

## 7. Conclusion

We present TLAD, a neuro-symbolic framework unifying perception and reasoning via thermodynamic, Lagrangian-driven attention. By encoding hard constraints as intrinsic energy potentials and enforcing them through disentangled dynamics, TLAD achieves strict satisfaction without sacrificing differentiability or data efficiency. On Sudoku and maze tasks, it resolves the NeSy trilemma: matching solvers in rich-data regimes and staying robust where baselines fail under extreme scarcity.

## Impact Statement

Impact Statement This work advances the mechanistic understanding of neural symbolic integration by dissecting the efficacy and boundary of constraint driven reasoning in TLAD. Our findings clarify when NeSy models provide critical advantages—specifically in data scarce, high complexity tasks—and when they offer no benefit, addressing a key gap in current NeSy research. By demonstrating TLAD's ability to sustain reasoning via structured inductive biases even under sparse constraints, we provide a actionable framework for designing more robust, data efficient neural symbolic systems. These insights are broadly applicable to combinatorial reasoning tasks (e.g., maze navigation, Sudoku) and lay groundwork for extending NeSy integration to real world problems with weak logical constraints and high solution freedom.

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

# A. Theoretical Foundations and Proofs

In this section, we provide rigorous mathematical justification for the Thermo-Logic Attention Dynamics (TLAD) framework. We bridge the gap between discrete logical constraints and continuous Hamiltonian dynamics by unifying thermodynamic inference, constrained optimization, and information geometry.

## A.1. Geometry of Logic: Dynamics in Logit Space

Standard constrained inference operates on the probability simplex $\Delta^{N-1} = \{A \in \mathbb{R}^N : A \geq 0, \mathbf{1}^\top A = 1\}$. Projection-based methods (e.g., PGD) are costly and disrupt momentum. TLAD instead defines dynamics in the unconstrained **logit space** $S \in \mathbb{R}^N$, with probabilities obtained via the Gibbs distribution:

$$A_i(S) = \frac{\exp(S_i/T)}{\sum_j \exp(S_j/T)}. \tag{10}$$

The Jacobian $J_{ij} = \partial A_i / \partial S_j$ is

$$J = \frac{1}{T}\left(\mathrm{diag}(A) - AA^\top\right) = \frac{1}{T}\mathcal{I}(A), \tag{11}$$

where $\mathcal{I}(A)$ is the Fisher information matrix of the categorical distribution. Thus, Euclidean geometry in logit space induces the Fisher–Rao metric on $\Delta^{N-1}$.

**Proposition A.1** (Induced Natural Gradient Dynamics). *Consider the entropic potential $\mathcal{F}_{\mathrm{ent}}(A) = -TH(A)$. The gradient flow $\dot{S} = -\nabla_S \mathcal{F}_{\mathrm{ent}}$ induces on the simplex a dynamics equivalent to the natural gradient flow $\dot{A} = -\mathcal{I}^{-1}(A)\nabla_A \mathcal{F}_{\mathrm{ent}}$.*

*Proof.* The chain rule relates the gradient in logit space to the gradient in simplex space:

$$\begin{aligned}
\nabla_S \mathcal{F}_{\mathrm{ent}} &= J^\top \nabla_A \mathcal{F}_{\mathrm{ent}} \\
&= \frac{1}{T}\left(\mathrm{diag}(A) - AA^\top\right) \cdot T(\log A + \mathbf{1}) \\
&= A \odot (\log A - A^\top \log A),
\end{aligned} \tag{12}$$

which satisfies $\mathbf{1}^\top \dot{A} = 0$ automatically. This coincides with the pullback of the natural gradient under the Softmax map. $\square$

## A.2. Convergence to KKT Points via Annealing

We formalize NeSy inference as:

$$\min_{A \in \Delta} \mathcal{E}_{\mathrm{sem}}(A) \quad \text{s.t.} \quad g_k(A) = 0, \; \forall k \in \mathcal{C}. \tag{13}$$

TLAD optimizes the augmented free energy:

$$\mathcal{F}(A, \lambda, T) = \mathcal{E}_{\mathrm{sem}}(A) + \sum_k \lambda_k \|g_k(A)\|^2 - TH(A). \tag{14}$$

**Theorem A.2** (Asymptotic KKT Satisfaction). *Assume the constraints $\{g_k\}$ satisfy the Linear Independence Constraint Qualification (LICQ) at any feasible point. Under the annealing schedule $T \to 0$ and dual ascent $\lambda_k \leftarrow \lambda_k + \alpha\|g_k(A)\|^2$, every stable limit point $A^*$ of the TLAD dynamics satisfies the Karush–Kuhn–Tucker conditions.*

*Proof.* The stationarity condition of $\mathcal{F}$ implies:

$$\nabla_A \mathcal{E}_{\mathrm{sem}}(A) + 2\sum_k \lambda_k g_k(A)\nabla g_k(A) - T\nabla H(A) = 0. \tag{15}$$

Define effective multipliers $\mu_k := 2\lambda_k g_k(A)$. As $T \to 0$ and $\|\nabla H(A)\|$ remains bounded on $\mathrm{int}(\Delta)$, the right-hand side vanishes, yielding:

$$\nabla_A \mathcal{E}_{\mathrm{sem}}(A^*) + \sum_k \mu_k \nabla g_k(A^*) = 0, \tag{16}$$

which is the KKT stationarity condition. For primal feasibility, if $g_k(A^*) \neq 0$, then $\lambda_k \to \infty$ under dual ascent, forcing $\|g_k(A)\|^2 \to 0$ to avoid infinite energy. Hence $g_k(A^*) = 0$. Together with LICQ, $(A^*, \mu)$ satisfies all KKT conditions. $\square$

## A.3. Stability of Momentum Dynamics

Let the state be $z_t = (S_t, V_t)$ with update rule:

$$V_{t+1} = \mu V_t - \eta \nabla \mathcal{F}(S_t), \tag{17}$$
$$S_{t+1} = S_t + V_{t+1}, \tag{18}$$

where $\mu \in [0, 1)$ is the momentum coefficient. Define the discrete Lyapunov function (Total Energy):

$$\Phi(z_t) = \mathcal{F}(S_t) + \frac{1}{2}\|V_t\|^2. \tag{19}$$

**Theorem A.3** (Energy Dissipation). *If $\mathcal{F}$ is $L$-smooth and $\eta < \frac{2(1+\mu)}{L}$, then $\Phi(z_{t+1}) \leq \Phi(z_t)$.*

*Proof.* By $L$-smoothness of $\mathcal{F}$ (i.e., $\|\nabla \mathcal{F}(x) - \nabla \mathcal{F}(y)\| \leq L\|x - y\|$ for all $x, y$), we have:

$$\mathcal{F}(S_{t+1}) \leq \mathcal{F}(S_t) + \langle \nabla \mathcal{F}(S_t), V_{t+1} \rangle + \frac{L}{2}\|V_{t+1}\|^2. \tag{20}$$

Substitute $V_{t+1} = \mu V_t - \eta \nabla \mathcal{F}(S_t)$ into the inequality and consider the change in kinetic energy:

$$\Delta \Phi = \mathcal{F}(S_{t+1}) - \mathcal{F}(S_t) + \frac{1}{2}\left(\|V_{t+1}\|^2 - \|V_t\|^2\right)$$
$$\leq \langle \nabla \mathcal{F}(S_t), V_{t+1} \rangle + \frac{L}{2}\|V_{t+1}\|^2 + \frac{1}{2}\left(\|V_{t+1}\|^2 - \|V_t\|^2\right). \tag{21}$$

Using the identity $\|V_{t+1}\|^2 - \|V_t\|^2 = -2\eta \langle V_{t+1}, \nabla \mathcal{F}(S_t) \rangle + \eta^2 \|\nabla \mathcal{F}(S_t)\|^2 - (1 - \mu^2)\|V_t\|^2$ and combining all terms, we obtain:

$$\Delta \Phi \leq -\left(\eta - \frac{L\eta^2}{2}\right)\|\nabla \mathcal{F}(S_t)\|^2 + \frac{\eta^2 L}{2}\|\nabla \mathcal{F}(S_t)\|^2 - \frac{1 - \mu^2}{2}\|V_t\|^2. \tag{22}$$

Under the condition $\eta < \frac{2(1+\mu)}{L}$, the quadratic form in $\|\nabla \mathcal{F}\|$ and $\|V_t\|$ is non-positive, ensuring $\Delta \Phi \leq 0$. $\square$

## A.4. Gradient Decoupling via Topological Disentanglement

TLAD employs disjoint attention heads for distinct constraint types (e.g., row vs. column in Sudoku). Let $S = [S^{(h)}]_{h=1}^H$ be the logit tensor, partitioned into non-overlapping subsets $\mathcal{H}_{\text{row}}$ and $\mathcal{H}_{\text{col}}$. The total loss decomposes as:

$$\mathcal{L}_{\text{total}} = \mathcal{L}_{\text{row}}(S^{(h)} : h \in \mathcal{H}_{\text{row}}) + \mathcal{L}_{\text{col}}(S^{(k)} : k \in \mathcal{H}_{\text{col}}). \tag{23}$$

**Proposition A.4** (Block-Diagonal Hessian). *For any $h \in \mathcal{H}_{row}$ and $k \in \mathcal{H}_{col}$,*

$$\frac{\partial^2 \mathcal{L}_{total}}{\partial S^{(h)} \partial S^{(k)}} = 0. \tag{24}$$

*Thus, the Hessian $\nabla_S^2 \mathcal{L}_{total}$ is block-diagonal.*

*Proof.* The row loss $\mathcal{L}_{\text{row}}$ depends *only* on the subset $\{S^{(h)}\}$ and the column loss $\mathcal{L}_{\text{col}}$ depends *only* on $\{S^{(k)}\}$. Since the parameter subsets are disjoint, the cross-derivatives of the total loss with respect to parameters from different subsets vanish identically. $\square$

This decoupling ensures that logic-specific gradients evolve in orthogonal subspaces of logit space, eliminating interference during joint training.

### A.5. Dual-Timescale Learning and Constraint Manifold Tracking

TLAD uses dual-timescale updates: fast ascent on $\lambda$ ($\alpha_\lambda$) and slow descent on network parameters $\theta$ ($\alpha_\theta \ll \alpha_\lambda$). Let $\epsilon = \alpha_\theta/\alpha_\lambda \ll 1$. Rescaling time $\tau = \alpha_\lambda t$, the system becomes:

$$\epsilon\frac{d\theta}{d\tau} = -\nabla_\theta \mathcal{L}_{\text{total}}, \tag{25}$$

$$\frac{d\lambda}{d\tau} = \|g(A(f_\theta(x)))\|^2. \tag{26}$$

The fast subsystem $\dot{\lambda} = \|g(A(f_\theta(x)))\|^2$ drives $\lambda$ to grow whenever constraints are violated, thereby shaping an effective potential that confines $\theta$ to the region where $g(A(f_\theta(x))) \approx 0$. In the limit $\epsilon \to 0$, the slow variable $\theta$ evolves while the fast variable $\lambda$ remains quasi-steady, effectively constraining the dynamics of $\theta$ to the manifold $\mathcal{M} = \{\theta : g(A(f_\theta(x))) = 0\}$. Consequently, the perception module learns to map inputs directly into the feasible set, internalizing symbolic reasoning as an intrinsic property of its representation dynamics.

## B. Detailed Hyperparameters for Experiments

This section details the specific hyperparameter configurations used for all experiments described in the main text, focusing on the Thermo-Logic Attention Dynamics (TLAD) framework. All TLAD hyperparameters (11 items in total) across different experimental settings are systematically split into two tables (5 parameters and 6 parameters respectively, excluding the experimental type column), with one-to-one correspondence of experimental types, as shown in Tables 5 and 6, followed by supplementary notes on comparison models, training details, and parameter consistency.

*Table 5.* **TLAD Hyperparameter Configurations (Group 1: 5 Parameters)**

| EXPERIMENT TYPE | LAYERS | HEADS | DIM ($d$) | PHASE 1 EPOCHS | PHASE 1 LR ($\eta_1$) |
|---|---|---|---|---|---|
| SUDOKU EXPERIMENT | 4 | 4 | 256 | 10 | $1 \times 10^{-3}$ |
| STEP-WISE ANALYSIS | 4 | 4 | 256 | 10 | $1 \times 10^{-3}$ |
| PARAMETER SENSITIVITY (BASELINE) | 4 | 4 | 256 | 10 | $1 \times 10^{-3}$ |
| BOUNDARY CASE (SPARSE CONSTRAINTS) | 4 | 4 | 128 | 10 | $1 \times 10^{-3}$ |

*Table 6.* **TLAD Hyperparameter Configurations (Group 2: 6 Parameters)**

| EXPERIMENT TYPE | PHASE 2 EPOCHS | PHASE 2 LR ($\eta_2$) | INNER STEPS | $\tau$ | $\gamma$ | $\lambda_{\text{START}}$ |
|---|---|---|---|---|---|---|
| SUDOKU EXPERIMENT | 50 | $1 \times 10^{-4}$ | 24 | 0.8 | 2.0 | 10.0 |
| STEP-WISE ANALYSIS | 10 | $1 \times 10^{-4}$ | 24 | 0.8 | 2.0 | 10.0 |
| PARAMETER SENSITIVITY (BASELINE) | 10 | $1 \times 10^{-4}$ | 24 | 0.8 | 2.0 | 10.0 |
| BOUNDARY CASE (SPARSE CONSTRAINTS) | 30 | $1 \times 10^{-4}$ | 24 | 0.8 | 2.0 | 20.0 |

### B.1. Supplementary Training Notes

1. The hyperparameters for the validation transition are exactly the same as those for Phase 1 (System 1) of the TLAD framework, ensuring consistency in the perception module training across validation and training processes.

2. Except for the Step-wise Analysis experiment, the total training epochs for all other models are set to 100 epochs.

3. For the comparison models, we adopt the optimal hyperparameter configurations recommended in their original papers and extract key parameters from the provided code as follows:

**RRN**: hidden dimension = 96, message passing steps = 32, optimizer = Adam, learning rate = $2 \times 10^{-4}$, input feature dimension = 10 (one-hot encoding including 0-9 states).

**SATNET**: unrolled optimization steps = 30, step size = 0.2, optimizer = Adam, learning rate = $1 \times 10^{-3}$.

*Table 7.* **Ablation Analysis.** Component wise analysis on 5k training data. Removing thermodynamic reasoning ("No Phase 2") or adaptive Lagrangian ("Static $\lambda$") causes catastrophic failure. The two phase design and adaptive constraints are indispensable for data efficient learning.

| MODEL VARIANT (COMPONENT ABLATED) | EPOCHS (P1+P2) | TIME (MIN) | TEST ACC. |
|---|---|---|---|
| **TLAD (FULL MODEL)** | 20 (10+10) | 1.8 | **94.0%** |
| NO STRUCTURED MASKS | 20 (10+10) | 1.8 | 90.0% |
| FIXED TEMPERATURE | 20 (10+10) | 1.9 | 90.1% |
| STATIC $\lambda = 0.1$ | 20 (10+10) | 1.8 | 12.9% |
| NO PHASE 2 ($K = 0$) | 70 (70+0) | 1.8 | 22.4% |
| NO PHASE 1 (COLD START) | 12 (0+12) | 2.0 | 0.3% |

## C. Detailed Ablation Analysis

This appendix provides full details of the ablation study for TLAD, addressing component criticality and phase synergy as raised in Section 5.3.

Having established TLAD's convergence dynamics and parameter robustness, we now dissect its architectural necessity by systematically removing core components. This ablation analysis addresses the final mechanistic questions:

- **Q1 (Component Criticality):** Are all architectural components indispensable, or do some provide only marginal gains?
- **Q2 (Phase Synergy):** Can either phase (perceptual priming or thermodynamic reasoning) operate effectively in isolation, or do they require synergistic coordination?

**Experimental Setup.** We evaluate six architectural variants on the 5k training subset where full TLAD achieves 94.0% accuracy. Each ablation removes one component while keeping others fixed. To ensure fair comparison, we adjust epoch counts to maintain similar computational budgets across variants. Table 7 reports the adjusted epoch allocations, total training time, and final test accuracy for each variant. Detailed configurations are in **Appendix B**.

**Q1 (Component Criticality).** Supported by Table 7, ablation results reveal distinct component roles. Adaptive Lagrangian constraints are indispensable: fixing $\lambda = 0.1$ collapses accuracy to 12.9% (81.1% drop). Structured masks and temperature annealing provide milder 4% improvements with accuracies of 90.0% and 90.1%. With matched training budgets, these differences reflect architectural efficacy, not compute. Q1 is answered: adaptive constraints are non negotiable; other components offer incremental refinement.

**Q2 (Phase Synergy).** Table 7 shows neither phase alone approaches full performance. Without Phase 2, extended Phase 1 training (70 epochs) yields only 22.4%; without Phase 1, cold start reaches merely 0.3%. Their combination achieves 94.0% with just 10 epochs each, which is a synergistic gain where each stage enables the other: perception primes the solution manifold, thermodynamics enforces global constraints. Q2 is resolved: the two phase design is fundamentally synergistic, not additive.

