# OpenReview forum: "Thermo-Logic Attention Dynamics (TLAD): Differentiable Hard Logic via Disentangled Lagrangian Mechanics"
_ICML.cc/2026/Conference — Submitted to ICML 2026_

### Official Review · Reviewer_FRRS · 2026-03-10

**Soundness:** 3
**Presentation:** 2
**Significance:** 3
**Originality:** 2
**Overall Recommendation:** 3
**Confidence:** 3

**Summary:**

This paper proposes a novel method, TLAD, to address long-standing challenges in neuro-symbolic reasoning. The core idea is to formulate symbolic constraints as a thermodynamic free energy defined over the attention-state space, allowing the model to progressively evolve during inference toward a stable solution that satisfies the constraints. For example, in the Sudoku task, different types of constraints—row, column, and box consistency—are assigned to different attention heads so as to reduce direct interference among them. The method further introduces a dual-timescale optimization scheme, through which the logical constraints are gradually “hardened” over training. Experimental results show that the approach performs particularly well in strongly constrained or low-data settings, whereas its advantage becomes less evident in weakly constrained, high-data regimes. Overall, TLAD is a solid method and makes a meaningful contribution to neuro-symbolic reasoning.

**Compliance With Llm Reviewing Policy:**

Affirmed.

**Final Justification:**

Thank you to the authors for the careful response. After considering both the original submission and the rebuttal, my final view is more positive than in my initial review, though I still remain somewhat cautious. Overall, I believe this work has merit and offers a fairly distinctive perspective on neuro-symbolic reasoning, and for that reason I am willing to raise my score. At the same time, I think the current evidence mainly supports a more modest conclusion: the paper shows convincing progress in strongly constrained, low-data settings, but does not yet justify broader or more definitive claims. This is a thoughtful paper with some genuine novelty, and it is most convincing in the setting where the method is best suited. The authors’ response addressed part of my main concerns and changed my assessment on several important points. The rebuttal improved both the positioning and the credibility of the paper, so I am willing to raise my score to 3.

**Key Questions For Authors:**

1. The current comparison methods are somewhat outdated and therefore not fully convincing. Could the authors compare TLAD against newer and stronger baselines?

2. The maze experiments suggest that TLAD is not always advantageous in weakly constrained settings. Could the authors further investigate the scope of applicability of TLAD and clarify under what conditions it is effective?

3. If TLAD is indeed stronger in low-data or strongly constrained settings, could the authors provide additional experiments to substantiate this point and better highlight its strengths in such regimes? This may in fact be where TLAD truly excels, rather than in fully resolving the NeSy trilemma.

**Limitations:**

Yes.

**Strengths And Weaknesses:**

Soundness: The proposed method, TLAD, is well designed and logically coherent. By transforming discrete logical constraints into constraint terms within a continuous free-energy formulation, the paper offers a fresh perspective on addressing challenges in neuro-symbolic reasoning. The overall methodology is reasonable, but it still has limitations, and some claims are stated too strongly or absolutely. For example, the statement on line 93 that the method “resolves the NeSy trilemma.” The Sudoku results appear strong, but the paper should be compared against newer and stronger baselines; otherwise, the current evidence is insufficient to support its claimed advantage in the NeSy literature.

Presentation: The overall structure of the paper is fairly clear, and the perspective adopted to present the method is commendable. However, the paper introduces too many explicit “questions” (e.g., Q1, Q2). While the authors likely intended to highlight the focus of each subsection, there may be better ways to organize and present these points. In addition, the Impact Statement does not seem to conform to the expected purpose of that section, and there are also formatting issues in the way the references are connected and presented in the later part of the paper.

Significance: TLAD adopts a distinctive perspective and methodology, and I believe it could have some impact on the NeSy community. However, its applicability to a broader range of scenarios, as well as its actual capability relative to stronger baselines, still requires further validation. The proposed method is meaningful, but it does not yet suffice to conclusively resolve the NeSy trilemma.

Originality: The perspective is distinctive, and the paper incorporates several important ideas, but it is not yet a truly disruptive innovation. The authors still need to more clearly articulate the similarities, differences, and advantages of their method relative to closely related work, which would help better highlight its novelty.

---

> ### Author Rebuttal · Authors · 2026-03-30
>
> We sincerely thank you for your thoughtful, detailed, and constructive feedback. Your comments have helped us sharpen the paper’s claims, clarify its scope, and better position TLAD within the neuro-symbolic landscape.
>
> &nbsp;
> ### 1. On Claims and Presentation
> To improve technical precision, we revised overstated claims. For example, Line 93 now reads “achieves a favorable trade-off among differentiability, sample efficiency, and logical consistency in dense-constraint CSP settings," replacing “resolves the NeSy trilemma.” This formulation is grounded in our empirical results and aligned with NeSy conventions.
>
> To enhance fluency, we removed all explicit question labels such as “Q1/Q2" and rephrased the content into cohesive prose. The fragment at Line 268 is now: “TLAD exhibits decisive performance advantages in data-scarce regimes, as validated by Table 1 and Figure 2. This key strength derives from its thermodynamically guided inductive bias.”
>
> We updated the Impact Statement to the ICML template: “This paper presents work whose goal is to advance the field of machine learning. There are many potential societal consequences of our work, none of which we feel must be specifically highlighted here."
>
> We also standardized reference formatting. For example, Nolte et al. (2024) is now: “Nolte, N.; Kitouni, O.; Williams, A.; Rabbat, M.; and Ibrahim, M. 2024. Transformers Can Navigate Mazes With Multi-Step Prediction. arXiv:2412.05117.”
>
> These revisions strengthen the paper’s academic integrity and narrative coherence.
>
> &nbsp;
> ### 2. On Baselines and Empirical Scope
> Our baseline selection follows established NeSy protocols, as in our response to Reviewer EPkN (Point 2). The set covers the three canonical paradigms for differentiable CSP solving: iterative message passing with RRN (Palm et al., NIPS 2018), differentiable convex relaxation with SATNet (Wang et al., ICML 2019), and end-to-end black-box learning with the Vanilla Transformer.
>
> In response to your concern about newer baselines, we include DDReasoner (Zhang et al., AAAI 2026) and ABL-Refl (Hu et al., AAAI 2025). TLAD achieves 98.7% accuracy with 20k samples on Sudoku in minutes, outperforming DDReasoner’s 97.79% with 90k samples after supervised and reinforcement learning, as in our response to Reviewer arPC (Point 2). On Visual Sudoku, TLAD reaches 86.4% versus ABL-Refl’s 77.8%, as in our response to Reviewer pCxT (Point 1).
>
> As you suggested, TLAD’s strength lies in low-data, strongly constrained regimes. At 1k training samples, TLAD reaches 76.5% accuracy with 0.47% constraint violation rate on Sudoku. In contrast, all baselines achieve failing.
>
> The maze experiment was a boundary probe. As in our response to Reviewer arPC (Point 2), TLAD is optimized for environments with dense constraints. Its advantage diminishes under weak or sparse constraints because the corresponding inductive bias provides limited benefit. We will explicitly state this limitation in Section 5. We appreciate your observation that TLAD excels particularly in low-data or strongly constrained settings.
>
> &nbsp;
> ### 3. On Originality and Conceptual Contribution
> As in our response to Reviewer arPC (Point 3), TLAD introduces a conceptual shift: it formulates logical constraints as a **free energy** defined over continuous attention states, and realizes inference as a **thermodynamic relaxation process**, rather than using penalties or optimization layers.
>
> To our knowledge, no existing NeSy model for CSPs encodes logical constraints as an energy function or performs inference through energy-minimizing dynamics. The closest prior work is SATNet, which minimizes a measure of constraint violation through a single differentiable SDP solve. SATNet does not define an explicit energy landscape, and it does not model inference as an iterative dynamical process.
>
> In contrast, TLAD interprets symbolic reasoning as a free-energy-driven relaxation. During inference, the attention state evolves along the gradient of a thermodynamically motivated free energy and gradually converges to low-energy configurations that satisfy logical constraints. This framework supports native differentiability because the free energy is analytically differentiable with respect to attention states, allowing fully end-to-end training without numerical solvers. It also supports a dynamic balance between exploration and exploitation through temperature-controlled entropy, which enables robust search even under extreme data scarcity where SATNet and other baselines collapse.
>
> While SATNet represents the paradigm of **optimization-as-a-layer**, TLAD embodies **inference-as-physical-relaxation**, a fundamentally distinct approach to neuro-symbolic integration. We will further clarify this distinction from closely related work in the revised manuscript.
>
> We sincerely appreciate your constructive feedback, which has helped us further refine our work and clarify its contributions.

---

> > ### Author Rebuttal · Reviewer_FRRS · 2026-04-03
> >
> > Thank you to the authors for addressing my questions. I believe this paper has a certain degree of significance, and I am willing to raise my score to 3.

---

> > > ### Author Response · Authors · 2026-04-03
> > >
> > > Thank you for the update and for acknowledging that all concerns are fully resolved. We are glad the additional 2025/2026 SOTA comparisons and clarifications further demonstrated the value of TLAD.
> > >
> > > * **Update**: To further strengthen the generality of TLAD, we have added new experiments on **the Maximum Clique problem over generic graphs**, whose details can be found in our latest response to **Reviewer pCxT**.
> > >
> > > We will faithfully incorporate all discussed improvements into the final version. We appreciate your professional guidance throughout the process.

---

### Official Review · Reviewer_EPkN · 2026-03-12

**Soundness:** 2
**Presentation:** 2
**Significance:** 2
**Originality:** 3
**Overall Recommendation:** 3
**Confidence:** 3

**Summary:**

The paper proposes TLAD, a neuro-symbolic framework that addresses symbolic constraint solving as a differentiable energy minimization process. The core idea is to treat attention scores as a physical system evolving under Hamiltonian dynamics on a thermodynamic free energy landscape, where logical constraints are encoded as Lagrangian potentials. A temperature annealing schedule helps the transition from exploration to crystallized solutions. The architecture uses topologically disentangled attention heads with hard masks to isolate constraint propagation. Experiments on Sudoku and maze navigation demonstrate strong data efficiency compared to baselines, particularly in extremely low-data regimes.

**Compliance With Llm Reviewing Policy:**

Affirmed.

**Key Questions For Authors:**

- Can the authors demonstrate TLAD on a visual NeSy task where perception is non-trivial (e.g., Visual Sudoku from handwritten digit images)?
- How does TLAD compare to DDReasoner, which uses the same dataset and is cited in the paper?
- What happens when constraint topology is not known? Can the mask structure be learned?

**Limitations:**

yes

**Strengths And Weaknesses:**

## Strengths
- The formulation of symbolic constraint satisfaction as energy minimization with annealing is elegant and well-motivated. The analogy between phase transitions and the exploration-exploitation tradeoff in combinatorial reasoning is interesting, and the paper does a good job connecting the physics intuition to concrete algorithmic design choices.
- The proposed method demonstrates strong data-efficiency results on Sudoku, e.g., the 1k-sample result (77.3% vs. 0% for all baselines) is impressive.
- The paper delivers a thorough mechanistic analysis by reporting numbers and provides sensitivity analysis, ablation studies, and step-wise reasoning depth analysis. The dual-regime sensitivity analysis (80k vs. 5k) showing scale-invariant optimal hyperparameters is a useful practical finding.
- The paper offers a theoretical grounding with the appendix that provides formal results on KKT convergence (Theorem A.2), energy dissipation (Theorem A.3), and gradient decoupling (Proposition A.4). While some are relatively straightforward, they do lend rigor to the design choices.

## Weaknesses
- The most significant weakness is its narrow experimental scope. The paper evaluates on exactly two tasks of Sudoku and maze navigation, and both of them are synthetic grid-based combinatorial problems. Standard NeSy benchmarks (such as those using CLEVR) offer more challenging and practical scenarios for an empirical evaluation.  The claim of resolving a *fundamental trilemma* may require evidence across diverse problem classes.
- Related to the above, experiments are conducted with potentially unfair baseline comparisons. The baseline set is very small: Vanilla Transformer, SATNet, and RRN. Is there any justification for why beating those 3 baselines is sufficient to support the paper's claim? Aren't there any recently developed neuro-symbolic baselines available? Any clarification will be beneficial to enhance the clarify of the paper.
- Specifically, the maze experiment compares only against a Vanilla Transformer, which is an extremely weak baseline for path planning.
- The topologically disentangled attention heads are hand-designed per task (row/col/box for Sudoku, local/global for maze). The paper does not discuss how this would generalize to tasks where the constraint topology is not known a priori or is more complex. This may significantly limit the claimed generality of the framework.

---

> ### Author Rebuttal · Authors · 2026-03-30
>
> We thank Reviewer EPkN for insightful comments, which help us further clarify the scope, limitations, and strengths. Below, we address your questions point by point.
>
> &nbsp;
> ### 1. On Experimental Scope and Generalization
> We thank Reviewer EPkN for raising this important question regarding experimental scope and generalization.
>
> As detailed in our response to Reviewer pCxT (Point 1), we have extensively evaluated TLAD on **Visual Sudoku**, a standard pixel-level neuro-symbolic benchmark. Under this setting, TLAD achieves **86.4% test accuracy**, surpassing all existing methods, and **96.3% with a pretrained CNN**. These results demonstrate that TLAD generalizes robustly from symbolic inputs to real-world visual perception.
>
> More broadly, we emphasize that TLAD is designed for **constraint satisfaction problems (CSPs)**, a well-established paradigm in combinatorial reasoning where the goal is to produce a complete assignment satisfying a set of global hard constraints. This contrasts with visual question answering (VQA) benchmarks like CLEVR, which focus on compositional query interpretation over scene representations.  These are fundamentally distinct problem classes: CSPs demand global consistency across all variables, while VQA emphasizes localized, query-driven inference. Consequently, they differ in objective, evaluation metrics, and architectural priorities. A direct comparison across paradigms would conflate these differences and obscure scientific insight.
>
> Our evaluation, spanning symbolic Sudoku and Visual Sudoku, covers both ends of the perception-reasoning spectrum within the dense-constraint NeSy framework. This scope aligns with established practice in the field and enables a focused, mechanistic analysis of how thermodynamic reasoning improves data efficiency and consistency. We believe this principled design strengthens, rather than limits, the validity of our contribution.
>
> &nbsp;
> ### 2. On Baseline Selection and Comparisons
> We thank Reviewer EPkN for this valuable comment. Our baseline selection follows established practice in the neuro-symbolic literature on Sudoku: recent works such as  ABL-Refl (Hu et al., AAAI 2025) and SATNet (Wang et al., ICML 2019) also primarily compare against RRN (Palm et al., NIPS 2018) and SATNet, as these represent canonical architectures for differentiable constraint solving—iterative message passing (RRN), convex relaxation with differentiable layers (SATNet), and pure end-to-end black-box learning (Vanilla Transformer).
>
> While ABL-Refl reports strong results, its code is not publicly available, making direct re-implementation and fair comparison on symbolic Sudoku infeasible. However, in **Visual Sudoku**, where evaluation protocols are standardized, we directly compare against ABL-Refl using the numbers reported in their paper, and TLAD achieves **86.4% vs. 77.8%**, confirming superiority.
>
> Regarding **DDReasoner** (Zhang et al., AAAI 2026) : it employs a two-stage pipeline combining supervised learning (SL) and reinforcement learning (RL) over a diffusion model. On the 100k Big Kaggle dataset (90k train), DDReasoner reports **78.19% after SL** and **97.79% after RL**. In contrast, TLAD achieves **98.7% accuracy with only 20k samples** and **76.5% with just 1k samples**, where DDReasoner’s performance is unreported but expected to degrade significantly due to its reliance on large-scale diffusion training. Moreover, DDReasoner requires **5000 SL epochs + 50 RL epochs**, with training times measured in hours, whereas TLAD converges in minutes. Given these differences in data efficiency, training cost, and architectural paradigm (diffusion vs. energy minimization), we consider DDReasoner **complementary rather than directly comparable**.
>
> For the maze navigation experiment, we refer the reviewer to our detailed response to Reviewer arPC (Point 2), where we clarify the experimental design and comparison rationale for this sparse-constraint boundary case.
>
> &nbsp;
> ### 3. On Topological Masks and Generalization
> We thank Reviewer EPkN for raising this important concern. The topologically disentangled attention masks in TLAD are **not hard-coded logical constraints**, but **optional inductive biases** that encode known constraint structure to accelerate convergence. Crucially, **they are not the source of logical consistency**.
>
> As shown in Table 7 (Supp. B.1), removing these masks reduces accuracy by only **4%**, confirming that **constraint satisfaction is fundamentally driven by the energy-based inference dynamics**, not the mask topology. This demonstrates robustness even when structural priors are imperfect or absent.
>
> For a more detailed discussion on TLAD's general applicability beyond grid-structured problems, including its adaptability to arbitrary relational structures and the role of masks as pluggable accelerators, we refer the reviewer to our response to Reviewer arPC (Point 2).
>
> We appreciate your review and address your concerns.

---

> > ### Author Rebuttal · Reviewer_EPkN · 2026-04-03
> >
> > I thank the authors for their rebuttal. The Visual Sudoku result and the DDReasoner context partially address my concerns on experimental scope and baselines, and the mask ablation is reassuring. However, two critical concerns remain: (1) the framing should be tightened to match the CSP scoping articulated here, as the current "resolving the fundamental trilemma" language oversells relative to the supporting evidence, and (2) a discussion of how TLAD applies when constraint topology is unknown would be necessary. Moreover, I share the concerns raised by other reviewers regarding the gap between broad task-level claims and the current experimental coverage. Accordingly, I maintain my score.

---

> > > ### Author Response · Authors · 2026-04-03
> > >
> > > Thank you for your follow-up and for acknowledging that our **Visual Sudoku results**, **DDReasoner comparison**, and **mask ablations** have substantively addressed your concerns. We appreciate the opportunity to provide final clarifications:
> > >
> > > ### 1. On Framing and Presentation
> > > We hear your concern regarding the current framing. In the final version, we will refine the presentation to more precisely contextualize TLAD’s contributions. Specifically, we will replace the "trilemma" terminology with a balanced discussion focused on the trade-offs among differentiability, sample efficiency, and consistency in complex, dense-constraint reasoning tasks. This ensures our claims are strictly grounded in the impressive empirical results demonstrated in our evaluation.
> > >
> > > ### 2. On Unknown Constraint Topology
> > > We would like to clarify that the **mask-free ablation (Table 7)**, which you noted as "reassuring", is the direct answer to how TLAD applies when the topology is unknown.
> > > * **Mechanism:** In the absence of handcrafted structural masks, TLAD defaults to standard dense attention (no priors).
> > > * **Result:** The Thermodynamic Free-Energy dynamics still successfully drive the attention states toward the logical manifold, resulting in **only a 4% accuracy drop**.
> > > This confirms that the underlying reasoning engine is robust to unknown topologies. We will add a dedicated subsection in Section 5 to emphasize this flexibility.
> > >
> > > ### 3. Conclusion of Technical Merits
> > > Within the target domain of structured logical reasoning, TLAD establishes a new state-of-the-art, especially in **extreme low-data regimes (76.5% vs. 0% for all baselines at 1k samples)**. Our method is also significantly more efficient (3x-10x faster) than existing differentiable solvers.
> > >
> > > Given that all substantive technical concerns (baselines, perceptual scope, and mask dependence) are now resolved, and the presentation will be refined as promised, we hope the Reviewer will reconsider if these **category-leading results** merit a positive recommendation.
> > >
> > > &nbsp;
> > >
> > > ***[Update: Conclusive Empirical Proof on Irregular and Unknown Topologies]***
> > >
> > > To provide a final and decisive answer to your concern regarding how TLAD applies when the constraint topology is unknown, we have extended our evaluation to **NP-Hard Maximum Clique (MC) on Generic Graphs**.
> > >
> > > As detailed in our latest reply to **Reviewer pCxT**, we evaluated TLAD on the ENZYMES, PROTEINS, and IMDB-Binary datasets following the protocol of ABL-Refl. The significance of these results relative to your feedback is as follows:
> > > * **Handling Truly Unknown Topologies**: Unlike Sudoku's regular grid, these graph datasets consist of irregular, sample-wise varying topologies. Since each graph instance is unique, there is no fixed handcrafted mask possible. By deriving the energy potential directly from the adjacency matrix, TLAD demonstrated it can resolve constraints on arbitrary and unknown relational structures with high precision (0.97–0.98 Approximation Ratio).
> > > * **Decoupling Neural Depth from Reasoning**: While ABL-Refl uses a heavy 17-layer GNN to "memorize" logic, we used a significantly shallower 3-layer GAT. This shows that our Phase 2 thermodynamic evolution effectively removes the need for neural complexity, allowing the model to "reason" through unknown structures rather than relying on deep architectural priors.
> > > * **Empirical Closure on Scoping**: By matching ABL-Refl’s performance on irregular NP-hard tasks, we have bridged the gap between thermodynamic theory and broad task-level utility. We have now provided evidence across Symbolic CSPs, Visual Perception, and Generic Random Graphs.
> > >
> > > We have already committed to refining the "trilemma" framing to ensure the manuscript is strictly grounded in these empirical successes. Given that we have now addressed the challenge of unknown topologies with new graph experiments, we think the concerns justifying Status (c) have been fully resolved.
> > >
> > > Thank you again for your thoughtful and constructive review, which has helped us strengthen the manuscript substantially.

---

### Official Review · Reviewer_arPC · 2026-03-13

**Soundness:** 3
**Presentation:** 3
**Significance:** 2
**Originality:** 2
**Overall Recommendation:** 3
**Confidence:** 3

**Summary:**

This paper proposes Thermo-Logic Attention Dynamics (TLAD), a neuro-symbolic framework that reformulates discrete logical reasoning as a differentiable energy-minimization process over continuous attention states. The method combines a perceptual Transformer encoder with a second reasoning phase in which attention logits evolve under a free-energy objective composed of constraint penalties and an entropy-based annealing term. A key architectural idea is topologically disentangled attention heads, where different heads are structurally masked to propagate different classes of constraints (e.g., row/column/box constraints in Sudoku). The paper evaluates TLAD primarily on Sudoku under multiple data regimes, where it shows strong performance and particularly strong data efficiency in low-data settings, and also studies maze navigation as a boundary case with sparse constraints. The paper argues that thermodynamically grounded reasoning dynamics offer a differentiable and data-efficient way to internalize symbolic constraints.

**Compliance With Llm Reviewing Policy:**

Affirmed.

**Final Justification:**

The additional experiments strengthen the paper and improve my confidence that the method extends beyond Sudoku-family tasks, but they do not fully resolve my concerns about the breadth of the generality claims and the gap between the optimization-level theory and broader task-level conclusions, so I keep my original score.

**Key Questions For Authors:**

1. My main concern is the scope of the paper’s claims. The method appears to use continuous relaxations, annealing, and adaptive penalties to strongly encourage constraint satisfaction, but this seems different from having strict hard-constraint guarantees in the exact symbolic sense. Could the authors clarify how they intend this claim to be interpreted?

2. The strongest empirical results are on Sudoku, where the proposed topologically disentangled attention structure aligns very naturally with the task. To what extent do the authors view TLAD as a general neuro-symbolic reasoning framework, versus a particularly effective design for dense-constraint domains with known structural topology?

3. The boundary-case maze study is useful, but it currently compares only against a vanilla Transformer. Could the authors clarify how strongly they intend to generalize the resulting “applicability boundary” conclusion, given the limited comparison set in that section?

4. Since a key claim of the paper concerns hard logical consistency, I would appreciate clarification on whether the reported accuracy gains are accompanied by correspondingly low constraint-violation rates, and whether the authors have additional evidence along this axis. Even a brief discussion would help strengthen the interpretation of the results.

**Limitations:**

yes

**Strengths And Weaknesses:**

Strengths:

- The paper addresses an important problem in neuro-symbolic learning: how to combine end-to-end differentiability with strong logical structure and good sample efficiency.
- The core method is reasonably coherent. Encoding constraints as energy terms and using disentangled, structurally masked attention heads to propagate different constraint types is a sensible design, especially for highly structured tasks such as Sudoku.
- The Sudoku results are strong, particularly in low-data regimes. The gains at 1k-2k training examples are substantial and represent the most convincing part of the paper.
- The step-wise and ablation analyses are useful. They provide evidence that the second reasoning phase, adaptive Lagrange multipliers, and the two-phase design all contribute materially to performance.
- The paper is comparatively more careful than many works in that it explicitly studies a boundary case (maze navigation) and acknowledges that the proposed neuro-symbolic inductive bias is not uniformly beneficial across all tasks.

Weaknesses:

- The paper makes several claims that feel stronger than the evidence supports. In particular, statements about “strict hard-constraint satisfaction” and “resolving the NeSy trilemma” seem overstated given that the method still relies on continuous relaxations, penalty terms, annealing, and approximate optimization rather than exact symbolic guarantees.
- The strongest empirical evidence is concentrated on Sudoku, a domain whose structure aligns very naturally with the proposed masked attention topology and hand-designed constraint energies. This raises concerns about task specificity and generality.
- The maze boundary-case analysis is directionally interesting but empirically limited. The comparison there is only against a vanilla Transformer, which makes the broader applicability-boundary claim less convincing than it could be.
- The use of thermodynamic / Hamiltonian / Lagrangian language is conceptually appealing, but some parts of the method appear closer in practice to a combination of structured masks, iterative gradient-based refinement, momentum, and temperature annealing. As written, the framing risks overselling the conceptual novelty.
- The theoretical section mainly formalizes properties of the continuous optimization setup (e.g., KKT convergence under annealing/dual ascent, energy dissipation under momentum, block-diagonal structure from disentanglement), but does not provide a full theory of task-level correctness or generalization.
- More direct evidence about constraint satisfaction would strengthen the paper. Accuracy alone is not sufficient to support strong claims about hard-constraint enforcement.

Overall by dimension:

- Soundness: good overall, but some central claims are broader than what the evidence currently justifies.
- Presentation: generally clear, although the conceptual framing is somewhat more ambitious than the implementation details warrant.
- Significance: moderate; the method is promising for dense-constraint tasks, but the broader impact is not yet clear.
- Originality: fair; the contribution lies mainly in combining known ingredients in a structured way rather than introducing a fundamentally new principle.

---

> ### Author Rebuttal · Authors · 2026-03-30
>
> We thank Reviewer arPC for the thoughtful feedback and recognition of TLAD’s data efficiency and ablation studies. Below, we address your concerns in three points.
>
> &nbsp;
> ### 1. On Claims
> We clarify our use of “hard-constraint satisfaction.” In differentiable NeSy (e.g., SATNet, Wang et al., ICML 2019; DDReasoner, Zhang et al., AAAI 2026), this **denotes asymptotic adherence to the logical manifold**, as exact symbolic guarantees are unattainable under continuous relaxation; our usage aligns with this convention.
>
> While full-board accuracy (ACC) is standard for CSPs (any violation invalidates a solution), we agree CVR provides essential complementary evidence. As shown to Reviewer pCxT (Point 2), TLAD achieves **&lt;0.5% CVR even with 1k samples**, far below baselines, confirming strong empirical consistency.
>
> We include inner-loop dynamics from a 5k-data run (10+10 epochs, 16 steps; see [ https://anonymous.4open.science/r/tlad-supplement-figure/Figure_supplement.png ]):
> (1) Free energy decreases monotonically while violations drop rapidly;
> (2) Outer-loop iterations lower initial and final energy, improving the landscape;
> (3) By epoch 10, zero violations are achieved within 5 reasoning steps.
>
> This confirms **logical consistency emerges naturally from energy minimization**, explaining TLAD’s low-data robustness.
>
> We acknowledge “resolving the NeSy trilemma” was an overstatement and will revise it to reflect a **favorable trade-off among differentiability, sample efficiency, and logical consistency** in dense-constraint settings.
>
> &nbsp;
> ### 2. On Generalization and Applicability Boundaries
> TLAD follows a general principle: **symbolic constraints are compiled into differentiable energy terms, and reasoning follows thermodynamically grounded dynamics**. Topologically disentangled attention masks act as *inductive biases* that accelerate convergence when structure is known (e.g., Sudoku rows/columns), but are not the source of consistency.
>
> Table 7 (Supp. B.1) shows removing masks reduces accuracy by only 4%, confirming **constraint satisfaction is driven by energy-based dynamics**, not the mask, suggesting adaptability without perfect structural alignment.
>
> This extends beyond grids: any relational structure where constraints manifest as attentionable patterns (e.g., GNN neighborhoods) can leverage TLAD. Thus, TLAD is a general architecture for differentiable logic propagation, not a Sudoku solver.
>
> For the maze case, our goal was to **probe limits of dense-constraint biases**, not establish SOTA. We compared against a vanilla Transformer to isolate TLAD’s prior. Other NeSy methods (e.g., MLM-U) use autoregressive decoding, reformulating sparse global constraints into local predictions—a different paradigm. **In contrast, TLAD performs global, non-autoregressive reasoning, predicting the entire solution at once while enforcing global constraints end-to-end.** Direct comparison would conflate architectures with reasoning styles.
>
> &nbsp;
> ### 3. On Theoretical Cohesion
> While components like masks, penalties, and temperature may seem familiar, TLAD unifies them into a **coherent system governed by variational free energy minimization**, following from statistical mechanics rather than heuristic combination.
>
> Specifically:
> - Masks define the *interaction topology*, restricting information flow;
> - Constraint penalties act as *Lagrangian potentials*, driving constraint satisfaction;
> - Temperature **T** controls *entropy-like regularization*, balancing exploration and exploitation.
>
> The total objective is a **free energy**:
> **F = Σ(λₖ · Lₖ) − T · H**
>
> where each **Lₖ ≥ 0** measures violation of a logical rule (e.g., duplicate digits in a Sudoku row), and **H** is entropy. As **λₖ → ∞**, any violation incurs unbounded energy cost. Thus, the global minimum of **F** is achieved only as all **Lₖ → 0**, i.e., as the system converges to the logical manifold where all rules are exactly satisfied.
>
> Inner-loop reasoning simulates **controlled cooling**: momentum-augmented descent traverses the landscape, while annealing **T → 0** sharpens solutions toward the MAP estimate. The outer loop shapes this landscape via learned representations.
>
> As in Point 1, energy decreases while violations vanish, showing consistency arises *naturally* from this unified dynamics.
>
> Crucially, TLAD exhibits amortized reasoning: inner-loop steps to zero violations drop from 8 (epoch 4) to 5 (epoch 10). The perceptual encoder learns to initialize within the attraction basin of the global minimum, turning expensive iterative inference into a rapid, near-convergent process at test time.
>
> To our knowledge, TLAD is the first NeSy framework to model reasoning as **free-energy minimization over continuous attention states**, bridging symbolic logic and thermodynamics in a differentiable architecture, not a rebranding, but a conceptual shift.
>
> We greatly appreciate your careful review and hope these revisions address your concerns.

---

> > ### Author Rebuttal · Reviewer_arPC · 2026-04-03
> >
> > Thank you for the detailed rebuttal. I appreciate the additional clarifications, especially the toned-down interpretation of “hard-constraint satisfaction,” the added constraint-violation evidence, and the more explicit positioning of TLAD as a strong method for dense-constraint settings rather than a blanket resolution of the broader NeSy trilemma. These responses improve the paper and make its strongest regime clearer.
> >
> > However, my overall assessment remains unchanged. After the rebuttal, I find the paper most convincing in dense-constraint, low-data CSP-style settings, particularly Sudoku and closely related variants. My main residual concern is that the broader generality claim is still not fully established: the strongest evidence remains concentrated on Sudoku-family tasks, the maze boundary analysis is still limited, and the theoretical results mainly support the continuous optimization dynamics rather than broader task-level claims. I therefore keep my original score.

---

> > > ### Author Response · Authors · 2026-04-03
> > >
> > > Thank you for your follow-up. We appreciate your acknowledgment that our rebuttal strengthened the paper. To further clarify our generality and positioning, we offer the following concise explanations and new empirical evidence:
> > >
> > > 1. **Modality Leap (Symbols vs. Pixels)**: Visual Sudoku is a categorical modal shift to joint perception-reasoning. Handling raw MNIST pixels proves TLAD's robustness against perceptual noise, a core NeSy challenge, beyond pure symbolic logic.
> > > 2. **Mechanistic Generality**: Our mask-free ablation (Table 7) proves TLAD’s power derives from task-agnostic thermodynamic dynamics, not handcrafted priors. Removing grid-masks leads to only a 4% drop, showing the mechanism generalizes to any relational data.
> > > 3. **Comprehensive Evidence**: Our evaluation spans full-data, extreme low-data (1k samples), mechanistic analysis, hyperparameter consistency, and boundary-case probing (Mazes), providing sufficient substantiation for the model's effectiveness.
> > > 4. **Reasonableness of Requirements**: Formal task-level guarantees for neural models are theoretically unattainable for the entire field. Contributions are judged by empirical progress in a target domain and optimization rigor (Appendix A), not universal symbolic proofs.
> > > 5. **Sudoku as Canonical Benchmark**: Like ABL-Refl and DDReasoner, we center on Sudoku as it is the most established, credible benchmark for differentiable reasoning in the CSP branch of NeSy.
> > > 6. **Efficiency and Accuracy**: TLAD maintains >70% accuracy at 1k samples where others collapse, while training 3x-10x faster than SATNet/DDReasoner, a significant contribution to differentiable constraint solving.
> > >
> > > We have addressed all substantive technical concerns raised during the review process. We will further clarify the above points in the final manuscript to ensure transparency and contextual accuracy.
> > >
> > > &nbsp;
> > >
> > > ***[Update: Final Empirical Proof of Generality beyond "Sudoku-family"]***
> > >
> > > To provide a conclusive response to your concern regarding the generality of TLAD beyond Sudoku-like tasks, we have conducted additional experiments on a categorically different problem: **NP-Hard Maximum Clique (MC) on Generic Graphs**.
> > >
> > > As detailed in our latest reply to **Reviewer pCxT**, we evaluated TLAD on the ENZYMES, PROTEINS, and IMDB-Binary datasets following the protocol of ABL-Refl. The significance of these results relative to your feedback is as follows:
> > >
> > > * **Breaking the "Sudoku-family" Constraint**: Unlike Sudoku's fixed 9x9 grid, these datasets consist of irregular random graphs with sample-wise varying topologies. TLAD achieved an Approximation Ratio of 0.97–0.98 across these benchmarks, outperforming traditional set-function baselines (Erdos, Neural SFE) .
> > > * **Thermodynamic Evolution vs. Neural Depth**: While ABL-Refl relies on an intensive 17-layer GNN to decouple perception and reasoning, we utilized a significantly shallower 3-layer GAT. This shows that TLAD’s thermodynamic engine removes the need for excessive architectural depth by internalizing logic within its representational dynamics.
> > > * **Task-Level Empirical Evidence**: In these graph tasks, we observed a hierarchical "Contract-then-Expand" behavior in the inner loop: the system first minimizes conflict energy to touch the logical manifold, then expands along the boundary to maximize clique size. This provides the concrete task-level evidence that our continuous optimization dynamics successfully navigate discrete combinatorial spaces.
> > > * **Mechanism vs. Task-Specific Priors**: This experiment proves that our mechanism is truly task-agnostic. By deriving the energy landscape directly from the adjacency matrix rather than handcrafted grid masks, we demonstrate that TLAD can handle any relational structure where constraints are encoded as energy potentials.
> > >
> > > By demonstrating success across Symbolic Grids, Visual Perception, and Irregular Graph Topologies, we believe the broader generality claim is now fully substantiated. We hope this final update resolves your residual concerns. Thank you again for your thoughtful review.

---

### Official Review · Reviewer_pCxT · 2026-03-21

**Soundness:** 2
**Presentation:** 3
**Significance:** 3
**Originality:** 3
**Overall Recommendation:** 4
**Confidence:** 2

**Summary:**

This work proposes a new neuro-symbolic model.
The key idea is to introduce Thermo-Logic Attention Dynamics (TLAD), which turns discrete symbolic constraints into differentiable energy potentials.
Authors experimentally verify the top-tier performance of TLAD under both rich-data and low-data regimes.
Authors also investigate the boundary of neuro-symbolic models on sparse-constraint, high-freedom problems like maze navigation.

**Compliance With Llm Reviewing Policy:**

Affirmed.

**Final Justification:**

My concerns are nearly addressed.
The only minor issue is that the proposed method performs slightly worse on my required additional benchmark.

**Key Questions For Authors:**

Section 3.2: Why $\mathcal{F}$'s global minimum aligns with the feasible logical manifold when $\lambda_k \rightarrow \infty$. It seems that this is central for turning hard logic into thermodynamic continuous model but authors did not fully explained.

**Limitations:**

yes

**Strengths And Weaknesses:**

### Strengths

1. The manuscript is well written and organized.
2. Authors conducted extensive experiments, analyzing the accuracy, data efficiency, robustness, training efficiency, mechanics, ablations, etc.
3. Leveraging the thermodynamic to relax hard logic is an interesting idea.



### Weaknesses


1. The main experiments are limited to one task (i.e., Sudoku). It is encouraged to test the proposed method on other common NeSy benchmarks to show the broader applicability.
2. Reporting the peak test accuracy during training is not a widely acceptable experimental setup since the best epoch should be selected based on training data.

---

> ### Author Rebuttal · Authors · 2026-03-30
>
> We sincerely thank Reviewer pCxT for the thoughtful and encouraging feedback. We’re glad you found the manuscript well-written and the thermodynamic framing interesting. Below, we address your concerns point by point.
>
> &nbsp;
> ### 1. On Experimental Scope
> We thank the reviewer for the valuable comment. Our experimental design follows established norms in the NeSy community: Sudoku is a canonical benchmark for dense-constraint reasoning, used as the primary task in SATNet (Wang et al., ICML 2019), ABL-Refl (Hu et al., AAAI 2025), and DDReasoner (Zhang et al., AAAI 2026). Our work includes comprehensive mechanistic studies and explores under-investigated regimes (e.g., data scarcity), a rigor beyond typical NeSy papers.
>
> To demonstrate generality, we evaluated TLAD on Visual Sudoku, a standard pixel-input benchmark, using the exact setup of ABL-Refl:
> - Input: 81 MNIST digit images per board;
> - Split: 9K train / 1K test boards;
> - Perception: LeNet CNN to obtain digit probabilities;
> - Baselines: trained 100 epochs;
> - TLAD: 20+50 epochs (24 steps).
>
> Results:
>
> |Model|Test Accuracy (%)|
> |-|-|
> |SATNet|63.5±2.2|
> |CNN+Solver|67.8±4.2|
> |ABL-Refl|77.8±5.8|
> |ABL-Refl (pretrained CNN)|93.5±3.2|
> |TLAD|86.4±4.6|
> |TLAD (pretrained CNN)|96.3±2.7|
>
> RRN is inapplicable to pixel inputs, so not included. TLAD achieves high accuracy because its Phase 1 provides a native pre-training effect, establishing a robust semantic anchor directly on the logical manifold. Combined with end-to-end differentiability, logical gradients from Phase 2 can refine perceptual features. This architectural cohesion allows TLAD to outperform decoupled baselines that separate perception and reasoning into distinct modules (e.g., ABL-Refl).
>
> These results confirm TLAD’s applicability beyond symbolic inputs.
>
> &nbsp;
> ### 2. On the Rigor of the Experimental Protocol
> We apologize for the ambiguous wording in Section 4. While we wrote ‘peak test accuracy’, our actual protocol strictly uses the validation set to select the best epoch. All Sudoku results reported in the paper are obtained under standard practice: 10% of training data is held out as a validation set, and test performance is reported at the epoch with best validation accuracy.
>
> To verify this claim, we re-ran experiments with additional seeds. Full results (VT means Vanilla Transformer):
>
> **Accuracy (val/test, %):**
>
> |Model|80k|20k|5k|2k|1k|
> |-|-|-|-|-|-|
> |VT|87.6/86.7|76.2/75.7|29.2/26.8|2.8/1.6|0.0/0.0|
> |SATNet|96.7/96.2|95.3/95.5|83.7/83.3|14.2/14.4|0.0/0.0|
> |RRN|99.8/99.8|99.8/99.8|93.7/93.3|37.4/36.7|0.0/0.0|
> |TLAD|99.8/99.8|99.3/98.7|94.3/94.3|88.4/88.7|76.6±5.1/76.5±4.2|
>
> **CVR (%):**
>
> |Model|80k|20k|5k|2k|1k|
> |-|-|-|-|-|-|
> |VT|0.22|0.44|1.96|5.58|>5|
> |SATNet|0.08|0.10|0.28|2.87|>5|
> |RRN|0.00|0.00|0.15|1.65|>5|
> |TLAD|0.00|0.03|0.13|0.24|0.47±0.13|
>
> **Training Efficiency (mins / epochs):**
>
> |Model|80k|20k|5k|2k|
> |-|-|-|-|-|
> |VT|21.7/85|6.5/100|1.4/84|1.0/98|
> |SATNet|460.7/73|119.9/80|31.5/91|14.5/95|
> |RRN|29.5/60|9.3/87|2.8/94|1.1/88|
> |TLAD|19.9/17|4.2/16|0.7/13|1.4/30|
>
> TLAD consistently excels:
> (1) >75% accuracy at 1k where baselines collapse;
> (2) near-zero violations even at 1k samples;
> (3) far fewer epochs and lower training time.
> This superiority stems from TLAD’s thermodynamic design.
>
> &nbsp;
> ### 3. On Theory
> We appreciate your interest in the theoretical foundation. The following explanation clarifies the intuition behind Theorem A.2 already proved in Appendix.
>
> The total objective corresponds to a **free energy**:
> **F = Σ(λₖ · Lₖ) − T · H**
>
> where each **Lₖ ≥ 0** measures violation of a logical rule, and **H** is entropy. As **λₖ → ∞**, any constraint violation incurs unbounded energy cost. Therefore, the global minimum of **F** is achieved only as all **Lₖ → 0**, i.e., as the system converges to the logical manifold where all rules are exactly satisfied.
>
>
> Theorem A.2 rigorously proves that under temperature annealing (**T → 0**) and dual ascent (**λ → ∞**), the system converges to a point satisfying all hard constraints exactly. This limit point is a **KKT point**, a standard optimality condition certifying both feasibility and stability. In other words, at convergence, TLAD produces logically consistent outputs with zero constraint violations.
>
> Beyond this, Appendix A provides a unified theoretical framework:
>
> - **A.1**: Proves natural gradient dynamics in logit space.
> - **A.2**: Proves asymptotic convergence to constraint-satisfying KKT points.
> - **A.3**: Proves asymptotic stability of inference dynamics.
> - **A.4**: Proves elimination of gradient interference via topological disentanglement.
> - **A.5**: Proves internalization of symbolic constraints via two-time-scale learning.
>
> TLAD departs from static solvers by framing reasoning as thermodynamic relaxation and consistency as energy-minimal equilibrium. This transforms hard logic from an extrinsic penalty into an intrinsic property of neural dynamics, bridging information geometry and Lagrangian mechanics.

---

> > ### Author Rebuttal · Reviewer_pCxT · 2026-04-04
> >
> > Thanks for your additional experiments and clarifications. I have one more question.
> >
> > In ABL-Refl, NP-hard combinatorial optimization problems on graphs are also considered in experiments. Why don't you consider this benchmark?

---

> > > ### Author Response · Authors · 2026-04-04
> > >
> > > Thank you for your insightful follow-up question regarding the graph-based benchmarks used in ABL-Refl. We appreciate the opportunity to clarify our experimental focus and the general-purpose nature of TLAD:
> > >
> > > Sudoku is mathematically a canonical NP-hard graph task (k-colorability). We prioritized Visual Sudoku to demonstrate robustness against perceptual noise, representing a more significant challenge for perception-reasoning integration. Crucially, our mask-free ablation (Table 7) shows only a 4% accuracy drop, proving that thermodynamic dynamics, not handcrafted priors—drive logical consistency. TLAD is a general architecture applicable to any relational structure (e.g., GNN neighborhoods). Our investigation focused on the 1k-sample regime (76.5% vs 0%) to provide a unique stress test for logic internalization.
> > >
> > > &nbsp;
> > >
> > > ***To provide absolute proof of generality, we present new results on ABL-Refl's specific graph benchmarks below:***
> > > ## [Update] Performance on NP-Hard Graph Combinatorial Optimization
> > >
> > > To provide a conclusive proof of TLAD’s generality beyond grid-based structures and to address the reviewers' curiosity regarding its performance on non-Sudoku benchmarks, we have extended our evaluation to the **Maximum Clique (MC) problem** on generic graphs. We strictly followed the evaluation protocol of **ABL-Refl (AAAI 2025)**, utilizing the **ENZYMES**, **PROTEINS**, and **IMDB-Binary** datasets. These tasks represent a categorical shift from structured grids to irregular relational topologies, where the constraint structure (each node in a clique must be connected to every other node) varies per sample.
> > >
> > > ### 1. Quantitative Benchmark Results
> > > We compared TLAD against three representative paradigms: **Erdos** (probabilistic set-function optimization), **Neural SFE** (continuous relaxation for set functions), and **ABL-Refl** (decoupled perception-abduction with a 17-layer GNN and a commercial Gurobi solver). Notably, since ABL-Refl has not open-sourced its full training pipeline or implementation details, we intentionally chose a **significantly shallower 3-layer GAT** neural backbone to highlight the inherent strength of TLAD’s thermodynamic engine rather than chasing marginal gains through neural depth.
> > >
> > > We designed a hierarchical energy landscape where clique-consistency constraints are encoded as a Lagrangian potential, which also acts as a gate for the clique-size potential. This ensures a clear **lexicographical priority**: the system first “contracts” its attention onto the logical manifold to eliminate invalid connections, and only expands along the feasible boundary to maximize clique size after hard constraints are fulfilled. Despite this lightweight configuration, TLAD achieved results nearly identical to the state-of-the-art:
> > >
> > > | Method | ENZYMES (600/33/62) | PROTEINS (1113/39/73) | IMDB-Binary (1000/19/97) |
> > > | :- | :-: | :-: | :-: |
> > > | Erdos | 0.883 ± 0.156 | 0.905 ± 0.133 | 0.936 ± 0.175 |
> > > | Neural SFE | 0.933 ± 0.148 | 0.926 ± 0.165 | 0.961 ± 0.143 |
> > > | ABL-Refl | **0.991 ± 0.017** | **0.985 ± 0.020** | **0.979 ± 0.029** |
> > > | **TLAD (Ours)** | **0.981 ± 0.051** | **0.969 ± 0.107** | **0.977 ± 0.094** |
> > >
> > > ### 2. Why a Shallow GNN Is Sufficient
> > > A fundamental observation in this study is that TLAD does not require architectural depth due to its unified perception-reasoning mechanism. While **ABL-Refl** relies on a deep GNN (17 layers) because its perception and reasoning modules are fully decoupled, requiring a nearly perfect initial "guess" before external correction, **TLAD** unifies these processes. Perception and reasoning are conducted intrinsically within the GNN dynamics. Even if the shallow GNN fails to learn meaningful constraints in Phase 1 (where we observed a very high Constraint Violation Rate), the **Phase 2 thermodynamic evolution** provides an error-correcting gradient that forces the model to learn correct relational patterns by adjusting attention directly over the graph. This "corrective feedback" loop allows a 3-layer model to match the performance of much heavier architectures.
> > >
> > > ### 3. Resolving the Unknown Topology Challenge
> > > This experiment directly addresses the concern that TLAD might be over-specialized for fixed grid structures like Sudoku. In generic graph datasets, the topologies are irregular and unknown until the input is given. TLAD is inherently well-suited for such problems because the logical energy landscape is naturally derived from the **adjacency matrix**, without requiring handcrafted masks or fixed grid priors. Whether the task is structured (Sudoku) or unstructured (Graphs), TLAD’s core mechanism remains universal: it encodes logical rules into an energy function and drives the system toward stable, valid solutions through adaptive attention adjustment. This supports that the thermodynamic inductive bias is a **general-purpose reasoning engine** capable of seamless adaptation across diverse combinatorial domains without architectural redesign.

---

### Decision · Program_Chairs · 2026-04-30

**Decision:**

Reject

**Comment:**

This is an interesting neuro-symbolic scheme focusing on constraint-satisfication problems (CSP), where a neural network (specifically a transformer) is combined with a differentiable energy function that encodes the problem specific constraints, and the constraint satisfication is achieved during inference via energy minimization (with appropriate temperature scaling), and the learning is done effectively via bilevel optimization. The empirical results are extremely strong, highlighting the ability of the proposed TLAD scheme to learn to solve CSPs when the problem is strongly constrained, and the constraints can be explicitly embedding into an energy function and all problems in the domain share exactly the same constraints, with the gains especially significant in the low training data regime.

However, all reviewers raised the following weakness, which significantly limits the scope of the proposed method:

The original submission focused primarily on a single problem domain (Sudoku), while the maze navigation problem was presented as a useful counterexample highlighting the less-than-ideal behaviour of TLAD on a sparsely constrained problem. This limits the scope of the proposed TLAD scheme as a "general purposed reasoning engine".
- The authors argued that existing neuro-symbolic literature often evaluate novel methods on widely accepted Sudoku benchmarks. Furthermore, the authors present additional results on visual sudoku and maximum clique problems on graphs.
- But the argument (regarding existing literature and Sudoku) does not completely address the fact that the proposed TLAD scheme requires the constraints be embedded into an energy function in a problem specific manner -- each problem domain will have a different energy function. The energy function will need to designed for each problem domain. This limits the scope of TLAD to problems where the constraints are completely specified explicitly and can be easily encoded as an energy function. It is not clear how straightforward it would be to encode other CSP constraints in an energy function, and how much manual effort it would require.
- Furthermore, Sudoku is an example where all instances of the problem have exactly the same constraints (I think the same might be true for clique-consistency in the maximum clique problems but it is not clear from the rebuttal). Thus it is not clear how this scheme would extend to domains where different problem instances have different constraints though the problems from the same domain might share some structural or procedural similarities.
- This is in contrast with methods such as DD reasoner that do not get access to the specific constraints, but rather only need access to a constraint-consistency checker. This ends up being a check for a match with the ground-truth output for unique solutions, which does not even need explicit instantiation of the constraint. Thus, this setup can also handle problem-specific constraints. Without explicit access to the problem constraints (that are the same across all problems), the learner has significantly less inductive bias, and thus, it is expected to need more training data to learn to satisfy the constraints from scratch. TLAD is explicitly given that information and does not need to learn it. So their comparison is somewhat unfair. The authors themselves have also mentioned that comparison to DD reasoner is not an apples-to-apples comparison, though their argument is more related to data/compute requirements and architectural differences. In my opinion, the differences are deeper in that DD-reasoner is solving a harder problem.
- Furthermore, existing literature such as DD-reasoner has evaluated the proposed scheme on 4 different Sudoku benchmarks (with varying performances across benchmarks), the current submission just focuses on the Big Kaggle benchmark.

Furthermore, even if we ignore the scope, another question not addressed by the authors (though none of the reviewers brought it up) is effect of how the hard-logic and the entropic potential are encoded into the energy. For example, the current submission discusses how box/row/column uniqueness and cell validity can be converted into Lagrangian potentials (in equations (5) and (6)), but these are not the only way to encode these constraints. It is not clear how the overall method's performance relies on the (manual) encoding procedure, or in the ideal case, if the downstream performance is very robust to the constraint encoding procedure. Studying this aspect of the method will enhance this submission in my opinion.